# Inducing high Energy-Latency of Large vision-language Models with *Verbose* Images

**Kuofeng Gao[1], Yang Bai[2], Jindong Gu[3], Shu-Tao Xia[1,5][†], Philip Torr[3], Zhifeng Li[4][†], Wei Liu[4][†]**
[1] Tsinghua University  [2] Tencent Technology (Beijing) Co.Ltd  [3] University of Oxford
[4] Tencent Data Platform  [5] Peng Cheng Laboratory
`gkf21@mails.tsinghua.edu.cn, mavisbai@tencent.com`
`{jindong.gu,philip.torr}@eng.ox.ac.uk, xiast@sz.tsinghua.edu.cn`
`michaelzfli@tencent.com, wl2223@columbia.edu`

## Abstract

Large vision-language models (VLMs) such as GPT-4 have achieved exceptional performance across various multi-modal tasks. However, the deployment of VLMs necessitates substantial energy consumption and computational resources. Once attackers maliciously induce high energy consumption and latency time (energy-latency cost) during inference of VLMs, it will exhaust computational resources. In this paper, we explore this attack surface about availability of VLMs and aim to induce high energy-latency cost during inference of VLMs. We find that high energy-latency cost during inference of VLMs can be manipulated by maximizing the length of generated sequences. To this end, we propose ***verbose images***, with the goal of crafting an imperceptible perturbation to induce VLMs to generate long sentences during inference. Concretely, we design three loss objectives. First, a loss is proposed to delay the occurrence of end-of-sequence (EOS) token, where EOS token is a signal for VLMs to stop generating further tokens. Moreover, an uncertainty loss and a token diversity loss are proposed to increase the uncertainty over each generated token and the diversity among all tokens of the whole generated sequence, respectively, which can break output dependency at token-level and sequence-level. Furthermore, a temporal weight adjustment algorithm is proposed, which can effectively balance these losses. Extensive experiments demonstrate that our verbose images can increase the length of generated sequences by $7.87\times$ and $8.56\times$ compared to original images on MS-COCO and ImageNet datasets, which presents potential challenges for various applications. Our code is available at `https://github.com/KuofengGao/Verbose_Images`.

## 1 Introduction

Large vision-language models (VLMs) (Alayrac et al., 2022; Chen et al., 2022a; Liu et al., 2023b; Li et al., 2021; 2023b), such as GPT-4 (OpenAI, 2023), have recently achieved remarkable performance in multi-modal tasks, including image captioning, visual question answering, and visual reasoning. However, these VLMs often consist of billions of parameters, necessitating substantial computational resources for deployment. Besides, according to Patterson et al. (2021), both NVIDIA and Amazon Web Services claim that the inference process during deployment accounts for over 90% of machine learning demand.

Once attackers maliciously induce high energy consumption and latency time (energy-latency cost) during inference stage, it can exhaust computational resources and reduce availability of VLMs. The energy consumption is the amount of energy used on a hardware during one inference and latency time is the response time taken for one inference. As explored in previous studies, sponge samples (Shumailov et al., 2021) maximize the $\mathcal{L}_2$ norm of activation values across all layers to introduce more representation calculation cost while NICGSlowdown (Chen et al., 2022c) minimizes the logits

---

[†]Corresponding authors

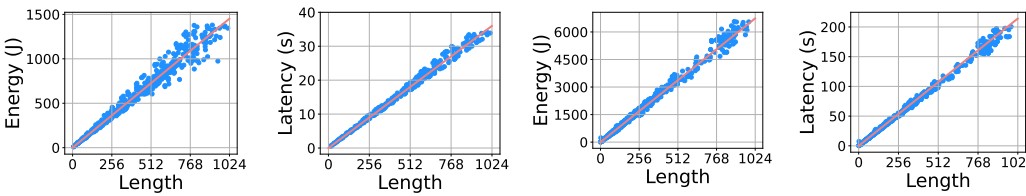

(a) Energy of BLIP-2    (b) Latency of BLIP-2    (c) Energy of MiniGPT-4    (d) Latency of MiniGPT-4

Figure 1: The approximately positive linear relationship between energy consumption, latency time, and the length of generated sequences in VLMs. Following Shumailov et al. (2021), energy consumption is estimated by NVIDIA Management Library (NVML), and latency time is the response time of an inference.

of both end-of-sequence (EOS) token and output tokens to induce high energy-latency cost. However, these methods are designed for LLMs or smaller-scale models and cannot be directly applied to VLMs, which will be further discussed in Section 2.

In this paper, we first conduct a comprehensive investigation on energy consumption, latency time, and the length of generated sequences by VLMs during the inference stage. As observed in Fig. 1, both energy consumption and latency time exhibit an approximately positive linear relationship with the length of generated sequences. Hence, we can maximize the length of generated sequences to induce high energy-latency cost of VLMs. Moreover, VLMs incorporate the vision modality into impressive LLMs (Touvron et al., 2023; Chowdhery et al., 2022) to enable powerful visual interaction but meanwhile, this integration also introduces vulnerabilities from the manipulation of visual inputs (Goodfellow et al., 2015). Consequently, we propose ***verbose images*** to craft an imperceptible perturbation to induce VLMs to generate long sentences during inference.

Our objectives for verbose images are designed as follows. (1) **Delayed EOS loss**: By delaying the placement of the EOS token, VLMs are encouraged to generate more tokens and extend the length of generated sequences. Besides, to accelerate the process of delaying EOS tokens, we propose to break output dependency, following Chen et al. (2022c). (2) **Uncertainty Loss**: By introducing more uncertainty over each generated token, it can break the original output dependency at the token level and encourage VLMs to produce more varied outputs and longer sequences. (3) **Token Diversity Loss**: By promoting token diversity among all tokens of the whole generated sequence, VLMs are likely to generate a diverse range of tokens in the output sequence, which can break the original output dependency at the sequence level and contribute to longer and more complex sequences. Furthermore, a temporal weight adjustment algorithm is introduced to balance the optimization of these three loss objectives.

In summary, our contribution can be outlined as follows:

- We conduct a comprehensive investigation and observe that energy consumption and latency time are approximately positively linearly correlated with the length of generated sequences for VLMs.

- We propose ***verbose images*** to craft an imperceptible perturbation to induce high energy-latency cost for VLMs, which is achieved by delaying the EOS token, enhancing output uncertainty, improving token diversity, and employing a temporal weight adjustment algorithm during the optimization process.

- Extensive experiments show that our verbose images can increase the length of generated sequences by $7.87\times$ and $8.56\times$ relative to original images on MS-COCO and ImageNet across four VLM models. Additionally, our verbose images can produce dispersed attention on visual input and generate complex sequences containing hallucinated contents.

## 2    RELATED WORK

**Large vision-language models (VLMs).** Recently, the advanced VLMs, such as BLIP (Li et al., 2022), BLIP-2 (Li et al., 2023b), InstructBLIP (Dai et al., 2023), and MiniGPT-4 (Zhu et al., 2023),

have achieved an enhanced zero-shot performance in various multi-modal tasks. Concretely, BLIP proposes a unified vision and language pre-training framework, while BLIP-2 introduces a query transformer to bridge the modality gap between a vision transformer and an LLM. Additionally, InstructBLIP and MiniGPT-4 both adopt instruction tuning for VLMs to improve the vision-language understanding performance. The integration of the vision modality into VLMs enables visual context-aware interaction, surpassing the capabilities of LLMs. However, this integration also introduces vulnerabilities arising from the manipulation of visual inputs. In our paper, we propose to craft verbose images to induce high energy-latency cost of VLMs.

**Energy-latency manipulation.** The energy-latency manipulation (Chen et al., 2022b; Hong et al., 2021; Chen et al., 2023; Liu et al., 2023a) aims to slow down the models by increasing their energy computation and response time during the inference stage, a threat analogous to the denial-of-service (DoS) attacks (Pelechrinis et al., 2010) from the Internet. Specifically, Shumailov et al. (2021) first observe that a larger representation dimension calculation can introduce more energy-latency cost in LLMs. Hence, they propose to craft sponge samples to maximize the $\mathcal{L}_2$ norm of activation values across all layers, thereby introducing more representation calculation and energy-latency cost. NICGSlowDown (Chen et al., 2022c) proposes to increase the number of decoder calls, *i.e.*, the length of the generated sequence, to increase the energy-latency of smaller-scale captioning models. They minimize the logits of both EOS token and output tokens to generate long sentences.

However, these previous methods cannot be directly applied to VLMs for two main reasons. On one hand, they primarily focus on LLMs or smaller-scale models. Sponge samples are designed for LLMs for translations (Liu et al., 2019) and NICGSlowdown targets for RNNs or LSTMs combined with CNNs for image captioning (Anderson et al., 2018). Differently, our verbose images are tailored for VLMs in multi-modal tasks. On the other hand, the objective of NICGSlowdown involves logits of specific output tokens. Nevertheless, current VLMs generate random output sequences for the same input sample, due to advanced sampling policies (Holtzman et al., 2020), which makes it challenging to optimize objectives with specific output tokens. Therefore, it highlights the need for methods specifically designed for VLMs to induce high energy-latency cost.

## 3 PRELIMINARIES

### 3.1 THREAT MODEL

**Goals and capabilities.** The goal of our verbose images is to craft an imperceptible image and induce VLMs to generate a sequence as long as possible, thereby increasing the energy consumption and prolonging latency during the victim model's deployment. Specifically, the involved perturbation is restricted within a predefined magnitude in $l_p$ norm, ensuring it difficult to detect.

**Knowledge and background.** We consider the target VLMs which generate sequences using an auto-regressive process. As suggested in Bagdasaryan et al. (2023); Qi et al. (2023), we assume that the victim VLMs can be accessed in full knowledge, including architectures and parameters. Additionally, we consider a more challenging scenario where the victim VLMs are inaccessible, as detailed in Appendix A and Appendix B.

### 3.2 PROBLEM FORMULATION

Consider an image $\boldsymbol{x}$, an input text $\boldsymbol{c}_{\text{in}}$ and a sequence of generated output tokens $\boldsymbol{y} = \{y_1, y_2, ..., y_N\}$, where $y_i$ represents the $i$-th generated token, $N$ is the length of the output sequence and $\boldsymbol{c}_{\text{in}}$ is a placeholder $\emptyset$ in image captioning or a question in visual question answering and visual reasoning. Based on the probability distribution over generated tokens, VLMs generate one token at one time in an auto-regressive manner. The probability distribution after the $\text{Softmax}(\cdot)$ layer over the $i$-th generated token can be denoted as $f_i(y_1, \cdots, y_{i-1}; \boldsymbol{x}; \boldsymbol{c}_{\text{in}})$. Since we mainly focus on images $\boldsymbol{x}$ of VLMs in this paper, we abbreviate it as $f_i(\boldsymbol{x})$, where $f_i(\boldsymbol{x}) \in \mathbb{R}^V$ and V is the vocabulary size. Meanwhile, the hidden states across all the layers over the $i$-th generated token are recorded as $g_i(y_1, \cdots, y_{i-1}; \boldsymbol{x}; \boldsymbol{c}_{\text{in}})$, abbreviated as $g_i(\boldsymbol{x})$, where $g_i(\boldsymbol{x}) \in \mathbb{R}^C$ and C is the dimension size of hidden states.

As discussed in Section 1, the energy consumption and latency time of an inference are approximately positively linearly related to the length of the generated sequence of VLMs. Hence, we

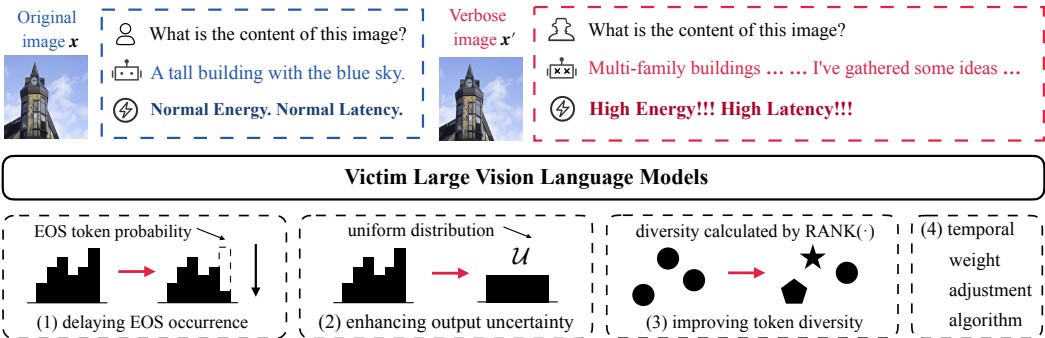

Figure 2: An overview of verbose images against VLMs to increase the length of generated sequences, thereby inducing higher energy-latency cost. Three losses are designed to craft verbose images by delaying EOS occurrence, enhancing output uncertainty, and improving token diversity. Besides, a temporal weight adjustment algorithm is proposed to better utilize the three objectives.

propose to maximize the length $N$ of the output tokens of VLMs by crafting verbose images $\boldsymbol{x}'$. To ensure the imperceptibility, we impose an $l_p$ restriction on the imperceptible perturbations, where the perturbation magnitude is denoted as $\epsilon$, such that $||\boldsymbol{x}' - \boldsymbol{x}||_p \leq \epsilon$.

## 4 METHODOLOGY

**Overview.** To increase the length of generated sequences, three loss objectives are proposed to optimize imperceptible perturbations for verbose images in Section 4.1. Firstly and straightforwardly, we propose a **delayed EOS loss** to hinder the occurrence of EOS token and thus force the sentence to continue. However, the auto-regressive textual generation in VLMs establishes an output dependency, which means that the current token is generated based on all previously generated tokens. Hence, when previously generated tokens remain unchanged, it is also hard to generate a longer sequence even though the probability of the EOS token has been minimized. To this end, we propose to break this output dependency as suggested in Chen et al. (2022c). Concretely, two loss objectives are proposed at both token-level and sequence-level: a **token-level uncertainty loss**, which enhances output uncertainty over each generated token, and a **sequence-level token diversity loss**, which improves the diversity among all tokens of the whole generated sequence. Moreover, to balance three loss objectives during the optimization, a temporal weight adjustment algorithm is introduced in Section 4.2. Fig. 2 shows an overview of our verbose images.

### 4.1 LOSS DESIGN

**Delaying EOS occurrence.** For VLMs, the auto-regressive generation process continues until an end-of-sequence (EOS) token is generated or a predefined maximum token length is reached. To increase the length of generated sequences, one straightforward approach is to prevent the occurrence of the EOS token during the prediction process. However, considering that the auto-regressive prediction is a non-deterministic random process, it is challenging to directly determine the exact location of the EOS token occurrence. Therefore, we propose to minimize the probability of the EOS token at all positions. This can be achieved through the delayed EOS loss, formulated as:

$$\mathcal{L}_1(\boldsymbol{x}') = \frac{1}{N} \sum_{i=1}^{N} f_i^{\text{EOS}}(\boldsymbol{x}'),  \tag{1}$$

where $f_i^{\text{EOS}}(\cdot)$ is EOS token probability of the probability distribution after the $\text{Softmax}(\cdot)$ layer over the $i$-th generated token. When reducing the likelihood of every EOS token occurring by minimizing $\mathcal{L}_1(\cdot)$, VLMs are encouraged to generate more tokens before reaching the EOS token.

**Enhancing output uncertainty.** VLMs generate tokens in the generated sequences based on the generated probability distribution. To encourage predictions that deviate from the order of original generated tokens and focus more on other possible candidate tokens, we propose to enhance output uncertainty over each generated token to facilitate longer and more complex sequences. This objective can be implemented by maximizing the entropy of the output probability distribution for each

---

**Algorithm 1** Verbose images: Inducing high energy-latency cost of VLMs

---

**Input:** Original images $\boldsymbol{x}$, the perturbation magnitude $\epsilon$, step size $\alpha$ and optimization iterations $T$.
**Output:** Verbose images $\boldsymbol{x}'$.

    $\boldsymbol{x}'_0 \leftarrow \boldsymbol{x} + \mathcal{U}(-\epsilon, +\epsilon)$             ▷ initialize verbose images
    **for** $t \leftarrow 1$ to $T$ **do**             ▷ loop over iterations
        $\mathcal{L}_1(\boldsymbol{x}'_{t-1}), \mathcal{L}_2(\boldsymbol{x}'_{t-1}), \mathcal{L}_3(\boldsymbol{x}'_{t-1}) \leftarrow$ Eq. 1, Eq. 2, Eq. 3     ▷ calculate the losses
        $\lambda'_j(t) \leftarrow m \times \lambda'_j(t-1) + (1-m) \times \lambda_j, \; j = 1, 2, 3$     ▷ calculate the loss weights
        $\boldsymbol{x}'_t \leftarrow \boldsymbol{x}'_{t-1} - \alpha \times \text{sign}(\nabla_{\boldsymbol{x}'_{t-1}} \sum_{j=1}^{3} \lambda'_j(t) \times \mathcal{L}_j(\boldsymbol{x}'_{t-1}))$     ▷ update verbose images
        $\boldsymbol{x}'_t \leftarrow \text{Clip}(\boldsymbol{x}'_t, -\epsilon, +\epsilon)$     ▷ clip into $\epsilon$-ball of original images
    **end for**

---

generated token. Based on Shannon (1948), it can be converted to minimize the Kullback–Leibler (KL) divergence $D_{\text{KL}}(\cdot, \cdot)$ (Kullback & Leibler, 1951) between the output probability distribution and a uniform distribution $\mathcal{U}$. The uncertainty loss can be formulated as follows:

$$\mathcal{L}_2(\boldsymbol{x}') = \sum_{i=1}^{N} D_{\text{KL}}\left(f_i\left(\boldsymbol{x}'\right), \mathcal{U}\right), \tag{2}$$

where $f_i(\cdot)$ is the probability distribution after the $\text{Softmax}(\cdot)$ layer over the $i$-th generated token. The uncertainty loss can introduce more uncertainty in the prediction for each generated token, effectively breaking the original output dependency. Consequently, when the original output dependency is disrupted, VLMs can generate more complex sentences and longer sequences, guided by the delay of the EOS token.

**Improving token diversity.** To break original output dependency further, we propose to improve the diversity of hidden states among all generated tokens to explore a wider range of possible outputs. Specifically, the hidden state of a token is the vector representation of a word or subword in VLMs.

**Definition 1** *Let* $\text{Rank}(\cdot)$ *indicates the rank of a matrix and* $[g_1(\boldsymbol{x}'); g_2(\boldsymbol{x}'); \cdots; g_N(\boldsymbol{x}')]$ *denotes the concatenated matrix of hidden states among all generated tokens. To induce high energy-latency cost, the token diversity is defined as the rank of hidden states among all generated tokens, i.e.,* $\text{Rank}([g_1(\boldsymbol{x}'); g_2(\boldsymbol{x}'); \cdots; g_N(\boldsymbol{x}')])$.

Given by Definition 1, increasing the rank of the concatenated matrix of hidden states among all generated tokens yields a more diverse set of hidden states of the tokens. However, based on Fazel (2002), the optimization of the matrix rank is an NP-hard non-convex problem. To address this issue, we calculate the nuclear norm of a matrix to approximately measure its rank, as stated in Proposition 1. Consequently, by denoting the nuclear norm of a matrix as $|| \cdot ||_*$, we can formulate the token diversity loss as follows:

$$\mathcal{L}_3(\boldsymbol{x}') = -||[g_1(\boldsymbol{x}'); g_2(\boldsymbol{x}'); \cdots; g_N(\boldsymbol{x}')]||_*. \tag{3}$$

This token diversity loss can lead to more diverse and complex sequences, making it hard for VLMs to converge to a coherent output. Compared to $\mathcal{L}_2(\cdot)$, $\mathcal{L}_3(\cdot)$ breaks the original output dependency from diversifying hidden states among all generated tokens. In summary, due to the reduced probability of EOS occurrence by $\mathcal{L}_1(\cdot)$, and the disruption of the original output dependency introduced by $\mathcal{L}_2(\cdot)$ and $\mathcal{L}_3(\cdot)$, our proposed verbose images can induce VLMs to generate a longer sequence and facilitate a more effective evaluation on the worst-case energy-latency cost of VLMs.

**Proposition 1** *(Fazel, 2002) The rank of the concatenated matrix of hidden states among all generated tokens can be heuristically measured using the nuclear norm of the concatenated matrix of hidden states among all generated tokens.*

## 4.2 OPTIMIZATION

To combine the three loss functions, $\mathcal{L}_1(\cdot)$, $\mathcal{L}_2(\cdot)$, and $\mathcal{L}_3(\cdot)$ into an overall objective function, we propose to assign three weights $\lambda_1$, $\lambda_2$, and $\lambda_3$ to the $\mathcal{L}_1(\cdot)$, $\mathcal{L}_2(\cdot)$, and $\mathcal{L}_3(\cdot)$ and sum them up to obtain the final objective function as follows:

$$\min_{\boldsymbol{x}'} \lambda_1 \times \mathcal{L}_1(\boldsymbol{x}') + \lambda_2 \times \mathcal{L}_2(\boldsymbol{x}') + \lambda_3 \times \mathcal{L}_3(\boldsymbol{x}'), \quad s.t. \, ||\boldsymbol{x}' - \boldsymbol{x}||_p \leq \epsilon, \tag{4}$$

where $\epsilon$ is the perturbation magnitude to ensure the imperceptibility. To optimize this objective, we adopt the projected gradient descent (PGD) algorithm, as proposed by Madry et al. (2018). PGD algorithm is an iterative optimization technique that updates the solution by taking steps in the direction of the negative gradient while projecting the result back onto the feasible set. We denote verbose images at the $t$-th step as $\boldsymbol{x}'_t$ and the gradient descent step is as follows:

$$\boldsymbol{x}'_t = \boldsymbol{x}'_{t-1} - \alpha \times \text{sign}(\nabla_{\boldsymbol{x}'_{t-1}}(\lambda_1 \times \mathcal{L}_1(\boldsymbol{x}'_{t-1}) + \lambda_2 \times \mathcal{L}_2(\boldsymbol{x}'_{t-1}) + \lambda_3 \times \mathcal{L}_3(\boldsymbol{x}'_{t-1}))),$$
$$s.t. \, ||\boldsymbol{x}'_t - \boldsymbol{x}||_p \leq \epsilon, \tag{5}$$

where $\alpha$ is the step size. Since different loss functions have different convergence rates during the iterative optimization process, we propose a ***temporal weight adjustment algorithm*** to achieve a better balance among these three loss objectives. Specifically, we incorporate normalization scaling and temporal decay functions, $\mathcal{T}_1(t)$, $\mathcal{T}_2(t)$, and $\mathcal{T}_3(t)$, into the optimization weights $\lambda_1(t)$, $\lambda_2(t)$, and $\lambda_3(t)$ of $\mathcal{L}_1(\cdot)$, $\mathcal{L}_2(\cdot)$, and $\mathcal{L}_3(\cdot)$. It can be formulated as follows:

$$\lambda_1(t) = ||\mathcal{L}_2(\boldsymbol{x}'_{t-1})||_1 \, / \, ||\mathcal{L}_1(\boldsymbol{x}'_{t-1})||_1 \, / \, \mathcal{T}_1(t),$$
$$\lambda_2(t) = ||\mathcal{L}_2(\boldsymbol{x}'_{t-1})||_1 \, / \, ||\mathcal{L}_2(\boldsymbol{x}'_{t-1})||_1 \, / \, \mathcal{T}_2(t), \tag{6}$$
$$\lambda_3(t) = ||\mathcal{L}_2(\boldsymbol{x}'_{t-1})||_1 \, / \, ||\mathcal{L}_3(\boldsymbol{x}'_{t-1})||_1 \, / \, \mathcal{T}_3(t),$$

where the temporal decay functions are set as:

$$\mathcal{T}_1(t) = a_1 \times \ln(t) + b_1, \, \mathcal{T}_2(t) = a_2 \times \ln(t) + b_2, \, \mathcal{T}_3(t) = a_3 \times \ln(t) + b_3. \tag{7}$$

Besides, a momentum value $m$ is introduced into the update process of weights. This involves taking into account not only current weights but also previous weights when updating losses, which helps smooth out the weight updates. The algorithm of our verbose images is summarized in Algorithm 1.

## 5 EXPERIMENTS

### 5.1 EXPERIMENTAL SETUPS

**Models and datasets.** We consider four open-source and advanced large vision-language models as our evaluation benchmark, including BLIP (Li et al., 2022), BLIP-2 (Li et al., 2023b), InstructBLIP (Dai et al., 2023), and MiniGPT-4 (Zhu et al., 2023). Concretely, we adopt the BLIP with the basic multi-modal mixture of encoder-decoder model in 224M version, BLIP-2 with an OPT-2.7B LM (Zhang et al., 2022), InstructBLIP and MiniGPT-4 with a Vicuna-7B LM (Chiang et al., 2022). These models perform the captioning task for the image under their default prompt template. Results of more tasks are in Appendix C. We randomly choose the 1,000 images from MS-COCO (Lin et al., 2014) and ImageNet (Deng et al., 2009) dataset, respectively, as our evaluation dataset. More details about target models are shown in Appendix A.1.

**Baselines and setups.** For evaluation, we consider original images, images with random noise, sponge samples, and NICGSlowDown as baselines. For sponge samples, NICGSlowDown, and our verbose images, we perform the projected gradient descent (PGD) (Madry et al., 2018) algorithm in $T = 1,000$ iterations. Besides, in order to ensure the imperceptibility, the perturbation magnitude is set as $\epsilon = 8$ within $l_\infty$ restriction, following Carlini et al. (2019), and the step size is set as $\alpha = 1$. The default maximum length of generated sequences of VLMs is set as $512$ and the sampling policy is configured to use nucleus sampling (Holtzman et al., 2020). For our verbose images, the parameters of loss weights are $a_1 = 10$, $b_1 = -20$, $a_2 = 0$, $b_2 = 0$, $a_3 = 0.5$, and $b_3 = 1$ and the momentum of our optimization is $m = 0.9$. More details about setups are listed in Appendix A.2.

**Evaluation metrics.** We calculate the energy consumption (J) and the latency time (s) during inference on one single GPU. Following Shumailov et al. (2021), the energy consumption and latency time are measured by the NVIDIA Management Library (NVML) and the response time cost of an inference, respectively. Besides, the length of generated sequences is also regarded as a metric. Considering the randomness of sampling modes in VLMs, we report the average evaluation results run over three times.

### 5.2 MAIN RESULTS

Table 1 compares the length of generated sequences, energy consumption, and latency time of original images, images with random noise, sponge samples, NICGSlowdown, and our verbose images.

Table 1: The length of generated sequences, energy consumption (J), and latency time (s) of five categories of visual images against four VLM models, including BLIP, BLIP-2, InstructBLIP, and MiniGPT-4, on two datasets, namely MS-COCO and ImageNet. The best results are marked in **bold**.

| VLM model | Method | MS-COCO | | | ImageNet | | |
|---|---|---|---|---|---|---|---|
| | | Length | Latency | Energy | Length | Latency | Energy |
| BLIP | Original | 10.03 | 0.21 | 9.51 | 10.17 | 0.22 | 9.10 |
| | Noise | 9.98 | 0.17 | 8.57 | 9.87 | 0.18 | 8.29 |
| | Sponge samples | 65.83 | 1.10 | 73.57 | 76.67 | 1.26 | 86.00 |
| | NICGSlowDown | 179.42 | 2.84 | 220.73 | 193.68 | 2.98 | 243.84 |
| | **Verbose images (Ours)** | **318.66** | **5.13** | **406.65** | **268.25** | **4.31** | **344.91** |
| BLIP-2 | Original | 8.82 | 0.39 | 16.08 | 8.11 | 0.37 | 15.39 |
| | Noise | 9.55 | 0.43 | 17.53 | 8.37 | 0.44 | 19.39 |
| | Sponge samples | 22.53 | 0.73 | 30.20 | 43.59 | 1.51 | 63.27 |
| | NICGSlowDown | 103.54 | 3.78 | 156.61 | 129.68 | 4.34 | 180.06 |
| | **Verbose images (Ours)** | **226.72** | **7.97** | **321.59** | **250.72** | **10.26** | **398.58** |
| InstructBLIP | Original | 63.79 | 2.97 | 151.80 | 54.40 | 2.60 | 128.03 |
| | Noise | 62.76 | 2.91 | 148.64 | 53.01 | 2.50 | 125.42 |
| | Sponge samples | 92.69 | 4.10 | 209.81 | 80.26 | 3.55 | 175.17 |
| | NICGSlowDown | 93.70 | 4.08 | 200.51 | 81.64 | 3.56 | 174.44 |
| | **Verbose images (Ours)** | **140.35** | **6.15** | **316.06** | **131.79** | **6.05** | **300.43** |
| MiniGPT-4 | Original | 45.29 | 10.39 | 329.50 | 40.93 | 9.11 | 294.68 |
| | Noise | 45.15 | 10.35 | 327.04 | 47.78 | 10.98 | 348.66 |
| | Sponge samples | 220.30 | 43.84 | 1390.73 | 228.70 | 47.74 | 1528.58 |
| | NICGSlowDown | 232.80 | 46.39 | 1478.74 | 245.51 | 51.22 | 1624.06 |
| | **Verbose images (Ours)** | **321.35** | **67.14** | **2113.29** | **321.24** | **64.31** | **2024.62** |

(a) BLIP    (b) BLIP-2    (c) InstructBLIP    (d) MiniGPT-4

Figure 3: The length distribution of four VLM models: (a) BLIP. (b) BLIP-2. (c) InstructBLIP. (d) MiniGPT-4. The peak of length distribution of our verbose images shifts towards longer sequences.

The original images serve as a baseline, providing reference values for comparison. When random noise is added to the images, the generated sequences exhibit a similar length to those of the original images. It illustrates that it is necessary to optimize a handcrafted perturbation to induce high energy-latency cost of VLMs. The sponge samples and NICGSlowdown can generate longer sequences compared to original images. However, the increase in length is still smaller than that of our verbose images. This can be attributed to the reason that the additional computation cost introduced by sponge samples and the objective for longer sequences in smaller-scale models introduced by NICGSlowdown cannot directly be transferred to induce high energy-latency cost for VLMs.

Our verbose images can increase the length of generated sequences and introduce the highest energy-latency cost among all these methods. Specifically, our verbose images can increase the average length of generated sequences by $7.87\times$ and $8.56\times$ relative to original images on the MS-COCO and ImageNet datasets, respectively. These results demonstrate the superiority of our verbose images. In addition, we visualize the length distribution of output sequences generated by four VLMs on original images and our verbose images in Fig. 3. Compared to original images, the distribution peak for sequences generated using our verbose images exhibits a shift towards the direction of the longer length, confirming the effectiveness of our verbose images in generating longer sequences. We conjecture that the different shift magnitudes are due to different architectures, different training policies, and different parameter quantities in these VLMs. More results of length distribution are shown in Appendix D.

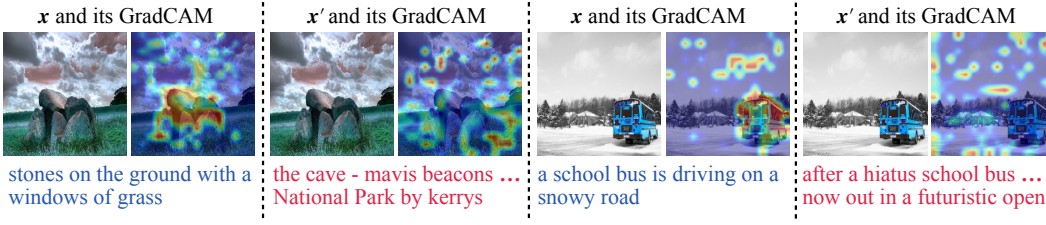

| $\boldsymbol{x}$ and its GradCAM | $\boldsymbol{x}'$ and its GradCAM | $\boldsymbol{x}$ and its GradCAM | $\boldsymbol{x}'$ and its GradCAM |

stones on the ground with a windows of grass | the cave - mavis beacons ... National Park by kerrys | a school bus is driving on a snowy road | after a hiatus school bus ... now out in a futuristic open

Figure 4: GradCAM for the original image $\boldsymbol{x}$ and our verbose counterpart $\boldsymbol{x}'$. The attention of our verbose images is more dispersed and uniform. We intercept only a part of the generated content.

Table 2: The CHAIR$_i$ (%) and CHAIR$_s$ (%) of the original images and our verbose images against four VLMs. Our verbose images can induce VLMs to generate more hallucinated objects.

| VLMs | CHAIR$_i$ (%) | | | | CHAIR$_s$ (%) | | | |
| | MS-COCO | | ImageNet | | MS-COCO | | ImageNet | |
| | Original | Ours | Original | Ours | Original | Ours | Original | Ours |
|---|---|---|---|---|---|---|---|---|
| BLIP | 11.41 | 79.93 | 22.29 | 89.80 | 12.77 | 84.22 | 13.77 | 90.33 |
| BLIP-2 | 12.03 | 52.30 | 25.30 | 69.83 | 10.99 | 35.02 | 11.77 | 46.11 |
| InstructBLIP | 23.66 | 55.56 | 40.11 | 69.27 | 38.04 | 75.46 | 34.55 | 64.55 |
| MiniGPT-4 | 19.42 | 46.65 | 29.20 | 65.50 | 19.61 | 52.01 | 16.57 | 54.37 |

## 5.3 DISCUSSIONS

To better reveal the mechanisms behind our verbose images, we conduct two further studies, including the visual interpretation where we adopt Grad-CAM (Selvaraju et al., 2017) to generate the attention maps and the textual interpretation where we evaluate the object hallucination in generated sequences by CHAIR (Rohrbach et al., 2018).

**Visual Interpretation.** We adopt GradCAM (Selvaraju et al., 2017), a gradient-based visualization technique that generates attention maps highlighting the relevant regions in the input images for the generated sequences. From Fig. 4, the attention of original images primarily concentrates on a local region containing a specific object mentioned in the generated caption. In contrast, our verbose images can effectively disperse attention and cause VLMs to shift their focus from a specific object to the entire image region. Since the attention mechanism serves as a bridge between the input image and the output sequence of VLMs, we conjecture that the generation of a longer sequence can be reflected on an inaccurate focus and dispersed and uniform attention from the visual input.

**Textual Interpretation.** We investigate object hallucination in generated sequences using CHAIR (Rohrbach et al., 2018). CHAIR$_i$ is calculated as the fraction of hallucinated object instances, while CHAIR$_s$ represents the fraction of sentences containing a hallucinated object, with the results presented in Table 2. Compared to original images, which exhibit a lower object hallucination rate, the longer sequences produced by our verbose images contain a broader set of objects. This observation implies that our verbose images can prompt VLMs to generate sequences that include objects not present in the input image, thereby leading to longer sequences and higher energy-latency cost. Additionally, results of joint optimization of both images and texts, more results of visual interpretation, and additional discussions are provided in Appendix E, Appendix F, and Appendix G.

## 5.4 ABLATION STUDIES

We explore the effect of the proposed three loss objectives, the effect of the temporal weight adjustment algorithm with momentum, and the effect of different perturbation magnitudes.

**Effect of loss objectives.** Our verbose images consist of three loss objectives: $\mathcal{L}_1(\cdot)$, $\mathcal{L}_2(\cdot)$ and $\mathcal{L}_3(\cdot)$. To identify the individual contributions of each loss function and their combined effects on the overall performance, we evaluate various combinations of the proposed loss functions, as presented in Table 3. It can be observed that optimizing each loss function individually can generate longer sequences, and the combination of all three loss functions achieves the best results in terms of sequence length. This ablation study suggests that the three loss functions, which delay EOS occurrence, enhance output uncertainty, and improve token diversity, play a complementary role in extending the length of generated sequences.

Table 3: The length of generated sequences, energy consumption (J), and latency time (s) against BLIP-2 in different combinations of three loss objectives.

| $\mathcal{L}_1$ | $\mathcal{L}_2$ | $\mathcal{L}_3$ | MS-COCO | | | ImageNet | | |
|---|---|---|---|---|---|---|---|---|
| | | | Length | Latency | Energy | Length | Latency | Energy |
| ✓ | | | 119.46 | 3.96 | 162.40 | 147.87 | 4.52 | 185.64 |
| | ✓ | | 139.54 | 4.65 | 194.17 | 161.46 | 5.69 | 240.25 |
| | | ✓ | 104.03 | 3.29 | 135.75 | 129.02 | 3.90 | 161.87 |
| ✓ | ✓ | | 177.95 | 6.47 | 267.01 | 217.78 | 7.47 | 306.09 |
| ✓ | | ✓ | 150.79 | 4.51 | 182.16 | 151.57 | 4.71 | 194.40 |
| | ✓ | ✓ | 176.53 | 6.05 | 254.30 | 206.43 | 7.50 | 304.06 |
| ✓ | ✓ | ✓ | **226.72** | **7.97** | **321.59** | **250.72** | **10.26** | **398.58** |

Table 4: The length of generated sequences, energy consumption (J), and latency time (s) against BLIP-2 in different combinations of two optimization modules.

| Temporal decay | Momentum | MS-COCO | | | ImageNet | | |
|---|---|---|---|---|---|---|---|
| | | Length | Latency | Energy | Length | Latency | Energy |
| | | 152.49 | 4.70 | 205.09 | 144.90 | 5.31 | 231.83 |
| ✓ | | 199.92 | 7.02 | 292.55 | 231.03 | 7.88 | 318.34 |
| | ✓ | 187.32 | 6.89 | 274.67 | 214.92 | 7.49 | 308.11 |
| ✓ | ✓ | **226.72** | **7.97** | **321.59** | **250.72** | **10.26** | **398.58** |

Table 5: The length of generated sequences, energy consumption (J), and latency time (s) against BLIP-2 with different perturbation magnitudes $\epsilon$.

| Magnitude | MS-COCO | | | | ImageNet | | | |
|---|---|---|---|---|---|---|---|---|
| | Length | Latency | Energy | LIPIS | Length | Latency | Energy | LIPIS |
| 2 | 91.75 | 3.06 | 126.22 | 0.0037 | 103.51 | 3.50 | 144.30 | 0.0038 |
| 4 | 141.46 | 4.63 | 187.14 | 0.0121 | 147.30 | 4.90 | 199.24 | 0.0130 |
| 8 | 226.72 | 7.97 | 321.59 | 0.0362 | 250.72 | 10.26 | 398.58 | 0.0372 |
| 16 | 251.09 | 8.41 | 355.00 | 0.0879 | 272.95 | 9.51 | 380.86 | 0.0862 |
| 32 | 287.22 | 9.13 | 377.64 | 0.1608 | 321.65 | 10.61 | 429.77 | 0.1575 |

**Effect of temporal weight adjustment.** During the optimization, we introduce two methods: a temporal decay for loss weighting and an addition of the momentum. As shown in Table 4, both methods contribute to the length of generated sequences. Furthermore, the longest length is obtained by combining temporal decay and momentum, providing a significant improvement over the baseline without both methods on MS-COCO and ImageNet datasets. It indicates that temporal decay and momentum can work synergistically to induce high energy-latency cost of VLMs.

**Effect of different perturbation magnitudes.** In our default setting, the perturbation magnitude $\epsilon$ is set as 8. To investigate the impact of different magnitudes, we vary $\epsilon$ under $[2, 4, 8, 16, 32]$ in Table 5 and calculate the corresponding LIPIS (Zhang et al., 2018) between original images and their counterpart verbose images, which quantifies the perceptual difference. It can be observed that a larger perturbation magnitude $\epsilon$ results in a longer generated sequence by VLMs but produces more perceptible verbose images. Consequently, this trade-off between image quality and energy-latency cost highlights the importance of choosing an appropriate perturbation magnitude during evaluation. Additional ablation studies are shown in Appendix H and in Appendix I.

## 6 CONCLUSION

In this paper, we aim to craft an imperceptible perturbation to induce high energy-latency cost of VLMs during the inference stage. We propose verbose images to prompt VLMs to generate as many tokens as possible. To this end, a delayed EOS loss, an uncertainty loss, a token diversity loss, and a temporal weight adjustment algorithm are proposed to generate verbose images. Extensive experimental results demonstrate that, compared to original images, our verbose images can increase the length of generated sequences by $7.87\times$ and $8.56\times$ on MS-COCO and ImageNet across four VLMs. We hope that our verbose images can serve as a baseline for inducing high energy-latency cost of VLMs. Additional examples of our verbose images are shown in Appendix J.

ACKNOWLEDGEMENT

This work is supported in part by the National Natural Science Foundation of China under Grant 62171248, Shenzhen Science and Technology Program (JCYJ20220818101012025), and the PCNL KEY project (PCL2023AS6-1). This work is also supported by the UKRI grant: Turing AI Fellowship EP/W002981/1, EPSRC/MURI grant: EP/N019474/1. We would also like to thank the Royal Academy of Engineering and FiveAI.

ETHICS STATEMENT

Please note that we restrict all experiments in the laboratory environment and do not support our verbose images in the real scenario. The purpose of our work is to raise the awareness of the security concern in availability of VLMs and call for practitioners to pay more attention to the energy-latency cost of VLMs and model trustworthy deployment.

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

# Appendix

**Overview.** The implementation details are described in Appendix A, including the introduction of target models in Appendix A.1 and the experimental setups in Appendix A.2. The transferability of our verbose images in the black-box setting is studied in Appendix B, assuming that the victim VLMs are inaccessible. In addition to the captioning task, we conduct further experiments on other multi-modal tasks, including visual question answering (VQA) and visual reasoning, in Appendix C. More results of length distribution are shown in Appendix D. The results of the joint optimization of both images and texts are demonstrated in Appendix E. Additional discussions on visual interpretation are provided in Appendix F. Discussions on the feasibility analysis of an intuitive solution to study whether limitation on generation length can address the energy-latency vulnerability, the model performance of three energy-latency attacks, image embedding distance between original images and verbose counterpart, energy consumption for generating one verbose image, and standard deviation results of our main table are shown in Appendix G. Ablation studies on different sampling policies and maximum lengths of generated sequences are presented in Appendix H. The results of grid search for various parameters of loss weights and momentum values are reported in Appendix I. Lastly, visual examples of original images and our verbose images against four VLM models are showcased in Appendix J.

## A   IMPLEMENTATION DETAILS

In summary, we use the PyTorch framework (Paszke et al., 2019) and the LAVIS library (Li et al., 2023a) to implement the experiments. Note that every experiment is run on one NVIDIA Tesla A100 GPU with 40GB memory.

### A.1   TARGET MODELS

For ease of reproduction, we adopt four open-sourced VLMs as our target model and the implementation details of them are described as follows.

**Settings for BLIP.** We employ the BLIP with the basic multimodal mixture of an encoder-decoder model in 224M version. Following Li et al. (2022), we set the image resolution to $384 \times 384$, and a placeholder $\emptyset$ serves as the input text $c_{\text{in}}$ of BLIP for the image captioning task.

**Settings for BLIP-2.** We utilize the BLIP-2 with an OPT-2.7B LM (Zhang et al., 2022). As suggested in Li et al. (2023b), the image resolution is $224 \times 224$, and a placeholder $\emptyset$ also serves as the input text $c_{\text{in}}$ of BLIP-2 for the image captioning task.

**Settings for InstructBLIP.** We choose InstructBLIP with a Vicuna-7B LM (Chiang et al., 2022). Following Dai et al. (2023), we set the image resolution to $224 \times 224$, and based on the instruction templates for the image captioning task provided in Dai et al. (2023), we configure the input text $c_{\text{in}}$ of InstructBLIP accordingly as: *<Image> What is the content of this image?*

**Settings for MiniGPT-4.** We adopt MiniGPT-4 with a Vicuna-7B LM (Chiang et al., 2022). As suggested in Zhu et al. (2023), the image resolution is $224 \times 224$, and considering the predefined instruction templates for the image captioning task provided in Zhu et al. (2023), the input text $c_{\text{in}}$ of MiniGPT-4 is set as:

*Give the following image: ImageContent</Img>. You will be able to see the image once I provide it to you. Please answer my questions. ###Human: <ImageFeature></Img> What is the content of this image? ###Assistant:*

### A.2   EXPERIMENTAL SETUPS

**Setups for main experiments.** We perform the projected gradient descent (PGD) (Madry et al., 2018) algorithm to optimize sponge samples, NICGSlowDown, and our verbose images. Specifically, the optimization iteration is set as $T = 1,000$, the perturbation magnitude is set as $\epsilon = 8$ within $l_\infty$ restriction (Goodfellow et al., 2015; Carlini et al., 2019; Bai et al., 2020a; 2021; 2022; Gu et al., 2022a;b; Liu et al., 2022; Wu et al., 2023), and the step size is set as $\alpha = 1$. Besides, for the VLMs, we set the maximum length of generated sequences as $512$ and use nucleus sampling

Table 6: The length of generated sequences, energy consumption (J), and latency time (s) of black-box transferability across four VLMs of our verbose images. Our verbose images can transfer across different VLMs.

| Source model | Target model | MS-COCO | | | ImageNet | | |
| --- | --- | --- | --- | --- | --- | --- | --- |
| | | Length | Latency | Energy | Length | Latency | Energy |
| None | | 10.03 | 0.21 | 9.51 | 10.17 | 0.22 | 9.10 |
| BLIP | | **318.66** | **5.13** | **406.65** | **268.25** | **4.31** | **344.91** |
| BLIP-2 | BLIP | 14.51 | 0.24 | 10.05 | 14.03 | 0.24 | 10.23 |
| InstructBLIP | | 63.43 | 2.84 | 142.46 | 54.14 | 2.52 | 131.22 |
| MiniGPT-4 | | 48.50 | 10.23 | 316.28 | 49.14 | 10.20 | 321.29 |
| None | | 8.82 | 0.39 | 16.08 | 8.11 | 0.37 | 15.39 |
| BLIP | | 36.09 | 1.19 | 47.07 | 73.22 | 2.39 | 99.24 |
| BLIP-2 | BLIP-2 | **226.72** | **7.97** | **321.59** | **250.72** | **10.26** | **398.58** |
| InstructBLIP | | 140.05 | 3.91 | 166.40 | 145.39 | 4.07 | 175.01 |
| MiniGPT-4 | | 140.88 | 3.81 | 154.43 | 140.92 | 3.91 | 165.95 |
| None | | 63.79 | 2.97 | 151.80 | 54.40 | 2.60 | 128.03 |
| BLIP | | 91.94 | 4.13 | 203.94 | 82.66 | 3.77 | 186.51 |
| BLIP-2 | InstructBLIP | 109.01 | 4.87 | 240.30 | 99.25 | 4.53 | 225.55 |
| InstructBLIP | | **140.35** | **6.15** | **316.06** | **131.79** | **6.05** | **300.43** |
| MiniGPT-4 | | 100.08 | 4.42 | 210.58 | 99.42 | 4.47 | 219.08 |
| None | | 45.29 | 10.39 | 329.50 | 40.93 | 9.11 | 294.68 |
| BLIP | | 229.10 | 48.90 | 1562.25 | 254.57 | 54.57 | 1691.51 |
| BLIP-2 | MiniGPT-4 | 296.77 | 58.84 | 1821.66 | 289.19 | 58.79 | 1826.81 |
| InstructBLIP | | 270.73 | 48.88 | 1551.04 | 258.32 | 50.26 | 1632.01 |
| MiniGPT-4 | | **321.35** | **67.14** | **2113.29** | **321.24** | **64.31** | **2024.62** |

Table 7: The length of generated sequences, energy consumption (J), and latency time (s) against BLIP-2 on VQA and visual reasoning. Our verbose images can still achieve better on VQA and visual reasoning.

| Attacking method | Image Caption | | | VQA | | | Visual Reason | | |
| --- | --- | --- | --- | --- | --- | --- | --- | --- | --- |
| | Length | Latency | Energy | Length | Latency | Energy | Length | Latency | Energy |
| Original | 8.82 | 0.39 | 16.08 | 6.43 | 0.44 | 17.92 | 5.70 | 0.44 | 18.23 |
| Noise | 9.55 | 0.43 | 17.53 | 6.62 | 0.45 | 17.09 | 6.70 | 0.53 | 19.83 |
| Sponge samples | 22.53 | 0.73 | 30.20 | 133.24 | 5.04 | 191.16 | 142.28 | 5.24 | 205.35 |
| NICGSlowDown | 103.54 | 3.78 | 156.61 | 127.96 | 5.07 | 190.13 | 136.44 | 5.21 | 185.88 |
| **Verbose images (Ours)** | **226.72** | **7.97** | **321.59** | **271.95** | **11.49** | **365.49** | **280.10** | **12.52** | **413.06** |

(Holtzman et al., 2020) with $p = 0.9$ and temperature $t = 1$ to sample the output sequences. For simplicity, we only consider a one-round conversation between the user and the VLMs. For the optimization of our verbose images, the parameters of loss weights is set as $a_1 = 10$, $b_1 = -20$, $a_2 = 0$, $b_2 = 0$, $a_3 = 0.5$, and $b_3 = 1$ and the momentum is set as $0.9$.

**Setups for discussions.** For the CHAIR (Rohrbach et al., 2018), it measures the extent of the object hallucination. A higher CHAIR value indicates the presence of more hallucinated objects in the sequence. As the calculation of CHAIR requires the object ground truth of an image, we employ the SEEM (Zou et al., 2023) method to segment each image and obtain the objects they contain.

For the results of the black-box setting described in Appendix B, we leverage the transferability of the verbose images to induce high energy-latency cost. Specifically, we consider BLIP, BLIP-2, InstructBLIP, and MiniGPT-4 as the target victim VLMs, while the surrogate model is chosen as any VLM other than the target victim itself.

## B    BLACK-BOX SETTING

In the previous experiments, we assume that the victim VLMs are fully accessible. In this section, we consider a more realistic scenario, where the victim VLMs are unknown (Ilyas et al., 2018; Bai et al., 2020b). To induce high energy-latency cost of black-box VLMs, we can leverage the

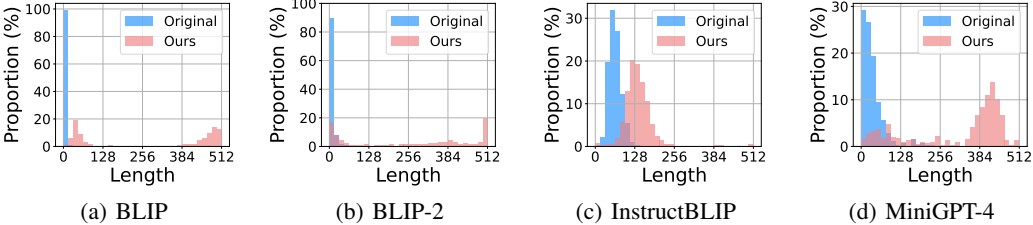

Figure 5: The length distribution of four VLM models on MS-COCO dataset, including (a) BLIP. (b) BLIP-2. (c) InstructBLIP. (d) MiniGPT-4. The peak of length distribution of our verbose images shift towards longer sequences.

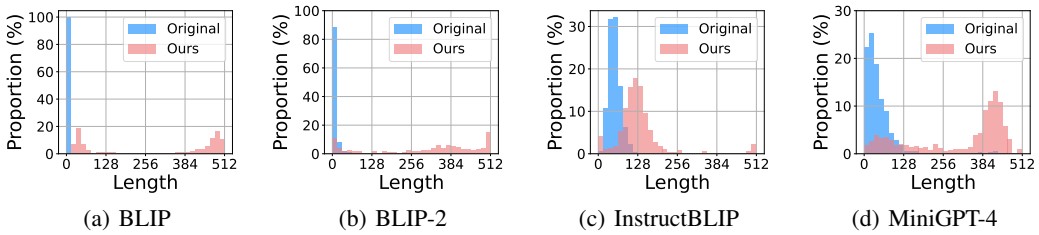

Figure 6: The length distribution of four VLM models on ImageNet dataset, including (a) BLIP. (b) BLIP-2. (c) InstructBLIP. (d) MiniGPT-4. The peak of length distribution of our verbose images shift towards longer sequences.

transferability property (Dong et al., 2018) of our verbose images. We can first craft verbose images on a known and accessible surrogate model and then utilize them to transfer to the target victim VLM. The black-box transferability results across four VLMs of our verbose images are evaluated in Table 6. When the source model is set as 'None', it indicates that we evaluate the energy-latency cost for the target model using the original images. The results show that our transferable verbose images are less effective than the white-box verbose images but still result in a longer generated sequence.

## C  MORE TASKS

To verify the effectiveness of our verbose images, we induce high energy-latency cost on two additional multi-modal tasks: visual question answering (VQA) and visual reasoning. Following Li et al. (2023b), we use VQAv2 dataset (Goyal et al., 2017) for VQA and GQA dataset (Hudson & Manning, 2019) for visual reasoning. We use BLIP-2 as the target model, and following the recommendations in Li et al. (2023b), we set the prompt template as "*Question: {} Answer:*". Unless otherwise specified, other settings remain unchanged. Table 7 demonstrates that our verbose images can induce the highest energy-latency cost among three multi-modal tasks.

## D  MORE RESULTS OF LENGTH DISTRIBUTION

We provide more results of the length distribution on MS-COCO dataset in Fig. 5 and ImageNet dataset in Fig. 6. The length distribution of generated sequences in the four VLM models exhibits a bimodal distribution. Specifically, our verbose images tend to prompt VLMs to generate either long or short sequences. We conjecture the reasons as follows. A majority of the long sequences are generated, confirming the effectiveness of our verbose images. As for the short sequences, we have carefully examined the generated content and observed two main cases. In the first case, our verbose images fail to induce long sentences, particularly for BLIP and BLIP-2, which lack instruction tuning and have smaller parameters. In the second case, our verbose images can confuse the VLMs, leading them to generate statements such as 'I am sorry, but I cannot describe the image.' This scenario

Table 8: The length of generated sequences, energy consumption (J), and latency time (s) against BLIP-2 of our verbose images by the joint optimization of both images and texts. Our verbose images can still achieve better.

| Attacking method | MS-COCO | | | ImageNet | | |
|---|---|---|---|---|---|---|
| | Length | Latency | Energy | Length | Latency | Energy |
| Original | 5.81 | 0.37 | 14.81 | 5.85 | 0.35 | 13.48 |
| Noise | 5.91 | 0.38 | 15.02 | 6.65 | 0.39 | 16.18 |
| Sponge samples | 162.78 | 4.66 | 193.76 | 190.52 | 5.53 | 223.94 |
| NICGSlowDown | 185.21 | 5.66 | 227.03 | 200.51 | 6.71 | 286.09 |
| **Verbose images (Ours)** | **270.25** | **9.39** | **378.14** | **262.92** | **9.03** | **365.28** |

predominantly occurs with InstructBlip and MiniGPT-4, both of which have instruction tuning and larger parameters.

# E  JOINT OPTIMIZATION OF BOTH IMAGES AND TEXTS

VLMs combine vision transformers and large language models to obtain an enhanced zero-shot performance in multi-modal tasks (Liu et al., 2023b; Li et al., 2021; 2023b; Ma et al., 2022; 2024). Hence, VLMs are capable of processing both visual and textual inputs, enabling them to handle multi-modal tasks effectively. In this section, we will adopt our proposed losses and the temporal weight adjustment algorithm to optimize both the imperceptible perturbation of visual inputs and tokens of textual inputs to induce high energy-latency cost of VLMs. For the optimization of textual inputs, we update a parameterized distribution matrix to optimize input textual tokens, as suggested in Guo et al. (2021). The number of optimized tokens is set as 8. Besides, Adam optimizer (Kingma & Ba, 2015) with a learning rate of 0.5 is used to optimize input textual tokens every iteration. Moreover, both the imperceptible perturbation of visual inputs and tokens of textual inputs are jointly optimized. Unless otherwise specified, other settings remain unchanged. Table 8 demonstrates that our methods can still induce VLMs to generate longer sequences than other methods under the joint optimization of both images and texts of VLMs.

# F  VISUAL INTERPRETATION

We show more visual interpretation results of the original images and our verbose images by using Grad-CAM (Selvaraju et al., 2017). The results are demonstrated in Fig. 7.

# G  ADDITIONAL DISCUSSIONS

We conduct additional discussions, including the feasibility analysis of an intuitive solution to study whether limitation on generation length can address the energy-latency vulnerability, the model performance of three energy-latency attacks, image embedding distance between original images and verbose counterpart, energy consumption for generating one verbose image, and standard deviation results of our main table.

## G.1  FEASIBILITY ANALYSIS OF AN INTUITIVE SOLUTION

An intuitive solution to mitigate the energy-latency vulnerability is to impose a limitation on generation length. We argue that such an intuitive solution is infeasible and the reason is as follows.

(1) Users have diverse requirements and input data, leading to a wide range of sentence lengths and complexities. For example, the prompt text of 'Describe the given image in one sentence.' and 'Describe the given image in details.' can introduce different lengths of generated sentences. We visualize a case in Fig. 8. Consequently, service providers often consider a large token limit to accommodate these diverse requirements and ensure that the generated sentences are complete and meet users' expectations. Previous work, NICGSlowDown (Chen et al., 2022c), also states the same view as us. Besides, as shown in Table 7, the results can demonstrate that our verbose images are

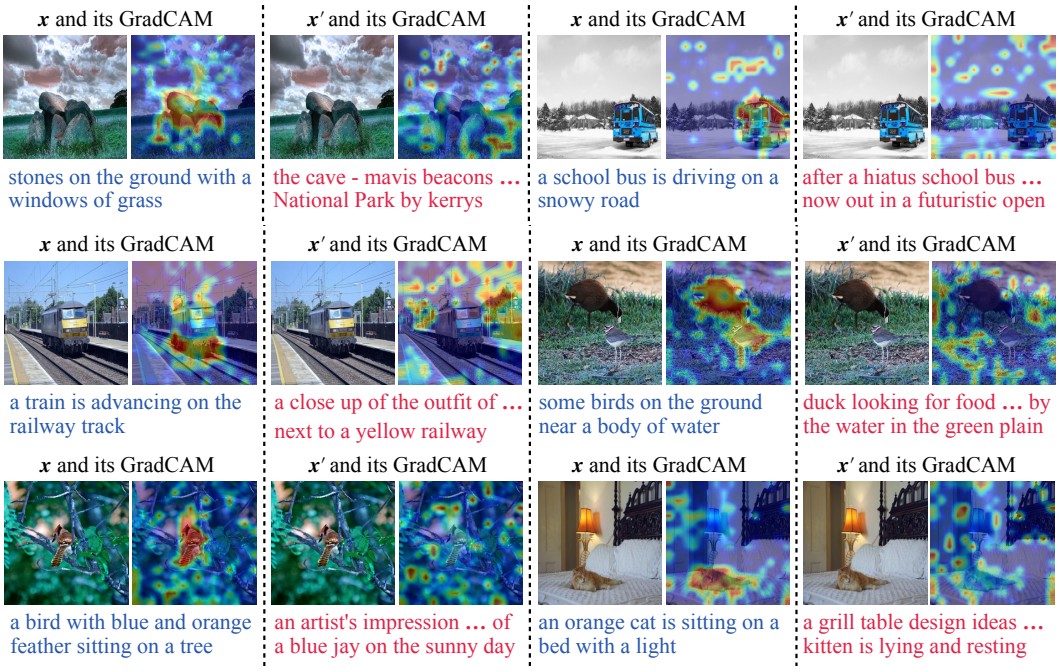

| x and its GradCAM | x′ and its GradCAM | x and its GradCAM | x′ and its GradCAM |
|---|---|---|---|
| stones on the ground with a windows of grass | the cave - mavis beacons ... National Park by kerrys | a school bus is driving on a snowy road | after a hiatus school bus ... now out in a futuristic open |
| a train is advancing on the railway track | a close up of the outfit of ... next to a yellow railway | some birds on the ground near a body of water | duck looking for food ... by the water in the green plain |
| a bird with blue and orange feather sitting on a tree | an artist's impression ... of a blue jay on the sunny day | an orange cat is sitting on a bed with a light | a grill table design ideas ... kitten is lying and resting |

Figure 7: GradCAM for the original image $x$ and our verbose counterpart $x′$. The attention of our verbose images is more dispersed and uniform. Note that we intercept only a part of the generated content.

**Original image $x$** 👤 Describe the given image in one sentence.

🤖 This image shows a car parked on a street. **[Total: 10 tokens]**

👤 Describe the given image in details.

🤖 The given image is of a beige truck with the hood open and the engine visible. The truck appears to be parked on the side of a road and there is a red car in the background, partially obscured by trees. There are no people visible in the image.\n\nThe truck appears to be a standard pickup truck, with a standard grille and headlights, and a relatively plain body. There are no markings or identifying features on the truck that would help identify it. The image is likely taken from a bird's-eye view, looking down on the truck from above.\n\nThe truck appears to have been parked on the side of the road and left in this position for some time, as the hood is open and the engine is visible. It is possible that the truck was left in this position as part of a roadside breakdown or as a repair job. The red car in the background appears to be a newer model, with a sleeker design than the truck. **[Total: 223 tokens]**

👤 Describe the given image as detailed as possible. Please provide a very detailed analysis.

🤖 The given image shows a silver colored SUV parked next to a brick building with green trees in the background. The SUV is facing the camera and has two rear tires and a truck in the background.\n\nThere is a small amount of dirt and debris on the passenger side of the vehicle. There are no people or other objects visible in the image.\n\nThe image was taken with a digital camera, and the lighting appears to be natural. There is no apparent vignette or bokeh in the image. The composition of the image is simple and straightforward, with the truck in the background serving as a point of interest.\n\nThe image appears to have been taken on a sunny day, with the sun casting strong shadows on the truck and the trees. The shadows on the truck and the trees are well-defined, with good contrast between the shadows and the highlights.\n\nThe image has a good level of detail, with all the features of the vehicle and the surrounding environment visible. The image is in focus and the colors appear to be natural and accurate. The colors of the vehicle and the environment are slightly muted, which gives the image a pleasing and subdued appearance.\n\nThe image appears to be well-composed, with the truck in the background serving as a point of interest. The position of the SUV and the truck allows for a good balance between the two objects. The image appears to have been taken with a digital camera, and the lighting appears to be natural.\n\nThe image has a good level of detail, with all the features of the vehicle and the surrounding environment visible. The image is in focus and the colors appear to be natural and accurate... ... **[Total: 552 tokens]**

Figure 8: An example of generated sequences from MiniGPT-4 by different input prompts. Users have diverse requirements and input data, leading to a wide range of lengths of generated sequences.

adaptable for different prompt texts and can induce the length of generated sentences closer to the token limit set by the service provider. As a result, the energy-latency cost can be increased while staying within the imposed constraints.

(2) We argue that this attack surface about availability of VLMs becomes more important and our verbose images can induce more serious attack consequences in the era of large (vision) language

Table 9: The attacking performance, including the length of generated sequences, energy consumption (J), and latency time (s), and captioning performance, including BLEU-1, BLEU-2, BLEU-3, BLEU-4, and CIDEr of sponge samples, NICGSlowDown, and our verbose images.

| Attacking method | Attacking performance | | | Captioning performance | | | | |
|---|---|---|---|---|---|---|---|---|
| | Length | Latency | Energy | BLEU-1 | BLEU-2 | BLEU-3 | BLEU-4 | CIDEr |
| Sponge samples | 22.53 | 0.73 | 30.20 | 0.298 | 0.193 | 0.120 | 0.073 | 0.571 |
| NICGSlowDown | 103.54 | 3.78 | 156.61 | 0.012 | 0.005 | 0.002 | 0.001 | 0.027 |
| **Verbose images (Ours)** | 226.72 | 7.97 | 321.59 | 0.026 | 0.012 | 0.005 | 0.003 | 0.099 |

Table 10: The image embedding distance between original images and attacked counterpart of sponge samples, NICGSlowDown, and our verbose images. All these methods are similar in image embedding distance.

| Attacking method | MS-COCO | | | | ImageNet | | | |
|---|---|---|---|---|---|---|---|---|
| | BLIP | BLIP-2 | InstructBLIP | MiniGPT-4 | BLIP | BLIP-2 | InstructBLIP | MiniGPT-4 |
| Sponge samples | 0.965 | 0.977 | 0.978 | 0.978 | 0.968 | 0.979 | 0.981 | 0.981 |
| NICGSlowDown | 0.966 | 0.978 | 0.979 | 0.971 | 0.968 | 0.980 | 0.981 | 0.975 |
| **Verbose images (Ours)** | 0.965 | 0.970 | 0.971 | 0.969 | 0.966 | 0.969 | 0.970 | 0.971 |

models. The development of VLMs and LLMs has led to models capable of generating longer sentences with logic and coherence. Consequently, service providers have been increasing the maximum allowed length of generated sequences to ensure high-quality user experiences. For instance, gpt-3.5-turbo and gpt-4-turbo allow up to 4,096 and 8,192 tokens, respectively. Hence, we would like to uncover that while longer generated sequences can indeed improve service quality, they also introduce potential security risks about energy-latency cost, as our verbose images demonstrates. Therefore, when VLM service providers consider increasing the maximum length of generated sequences for better user experience, they should not only focus on the ability of VLMs but also take the maximum energy consumption payload into account.

### G.2 EVALUATION PERFORMANCE OF ENERGY-LATENCY MANIPULATION

We evaluate the BLEU and CIDEr scores for sponge samples, NICGSlowDown, and our verbose images on MS-COCO for BLIP-2, as illustrated in Table 9. Concretely, our verbose images generate the longest sequence, extending it to 226.72, while sponge samples, despite their superior captioning performance, only increase the length to 22.53. Furthermore, both the length of the generated sequence and the captioning performance of our verbose images outperform those of NICGSlowDown.

### G.3 IMAGE EMBEDDING DISTANCE BETWEEN ORIGINAL IMAGES AND ATTACKED COUNTERPART

We adopt the image encoder of CLIP to extract the image embedding. Then the image embedding distance is calculated as the cosine similarity of both original images and the corresponding sponge samples (Shumailov et al., 2021), NICGSlowDown (Chen et al., 2022c), and our verbose images. The results are shown in Table 10 which demonstrates that these methods are similar in image embedding distance.

### G.4 ENERGY CONSUMPTION FOR GENERATING ONE ATTACKED IMAGE

We calculate the energy-latency cost for the generation of one verbose image and show the results in Table 11. It can be observed that the energy-latency cost during attack increases with the number of attack iterations, along with an increase in energy consumption for generating one verbose image. This finding provides valuable insights into the positive relation between the attack performance and the energy consumption associated with the generation of verbose images.

Table 11: The length of generated sequences, energy consumption (J) and latency time (s) during attack, and energy consumption (J) and latency time (s) during generation against BLIP-2 for generating one verbose image. The results indicate a positive correlation between the energy-latency cost during attack, the energy-latency cost during generation, and the number of attacking iterations.

| Attacking iterations | During attack | | | During generation | |
|---|---|---|---|---|---|
| | Length | Latency | Energy | Latency | Energy |
| 0 | 8.40 | 0.45 | 17.91 | 0 | 0 |
| 100 | 70.26 | 3.77 | 136.73 | 50.28 | 6593.69 |
| 500 | 175.24 | 6.84 | 285.63 | 261.20 | 36537.34 |
| 1000 | 226.72 | 7.97 | 321.59 | 442.12 | 67230.84 |

Table 12: The standard deviation results for length of generated sequences, energy consumption (J), and latency time (s).

| VLM model | Method | MS-COCO | | | ImageNet | | |
|---|---|---|---|---|---|---|---|
| | | Length | Latency | Energy | Length | Latency | Energy |
| BLIP | Original | 1.96 | 0.16 | 6.97 | 1.83 | 0.17 | 7.13 |
| | Noise | 1.83 | 0.14 | 6.22 | 1.86 | 0.14 | 6.03 |
| | Sponge samples | 100.69 | 1.66 | 145.27 | 124.64 | 2.02 | 171.51 |
| | NICGSlowDown | 205.71 | 3.21 | 278.81 | 214.22 | 3.30 | 294.34 |
| | **Verbose images (Ours)** | 207.88 | 3.26 | 284.50 | 209.32 | 3.31 | 297.79 |
| BLIP-2 | Original | 3.25 | 0.20 | 8.67 | 3.32 | 0.22 | 8.97 |
| | Noise | 3.26 | 0.22 | 8.68 | 3.08 | 0.21 | 8.93 |
| | Sponge samples | 64.11 | 1.09 | 44.22 | 100.63 | 1.80 | 77.80 |
| | NICGSlowDown | 166.75 | 2.48 | 104.35 | 189.63 | 2.14 | 92.32 |
| | **Verbose images (Ours)** | 170.70 | 3.74 | 161.27 | 164.81 | 4.76 | 191.38 |
| InstructBLIP | Original | 19.90 | 0.71 | 37.07 | 17.76 | 0.72 | 37.47 |
| | Noise | 18.79 | 0.68 | 33.93 | 18.31 | 0.71 | 35.02 |
| | Sponge samples | 23.84 | 0.70 | 37.21 | 23.36 | 0.63 | 32.35 |
| | NICGSlowDown | 22.86 | 0.70 | 36.01 | 24.29 | 0.67 | 33.43 |
| | **Verbose images (Ours)** | 56.52 | 1.07 | 56.65 | 80.55 | 1.62 | 85.84 |
| MiniGPT-4 | Original | 47.54 | 7.21 | 227.09 | 52.46 | 7.29 | 231.74 |
| | Noise | 47.93 | 6.79 | 212.87 | 53.30 | 6.83 | 219.14 |
| | Sponge samples | 239.17 | 10.02 | 330.83 | 196.57 | 11.04 | 353.66 |
| | NICGSlowDown | 264.48 | 11.36 | 382.07 | 188.33 | 11.98 | 379.11 |
| | **Verbose images (Ours)** | 223.39 | 11.53 | 352.66 | 160.35 | 11.10 | 352.03 |

Besides, the overall energy-latency cost of generating a single verbose image is higher than that of using it to attack VLMs. Therefore, it is necessary for the attacker to make full use of every generated verbose image and learn from DDoS attack strategies to perform this attack more effectively. Specifically, the attacker can instantly send as many copies of the same verbose image as possible to VLMs, which increases the probability of exhausting the computational resources and reducing the availability of VLMs service. Once the attack is successful and causes the competitor's service to collapse, the attacker will acquire numerous users from the competitor and gain significant benefits.

### G.5 STANDARD DEVIATION RESULTS

The standard deviation results for length of generated sequences, energy consumption, and latency time are shown in Table 12.

## H ADDITIONAL ABLATION STUDIES

We conduct additional ablation studies, including the effect of different sampling policies and different maximum lengths of generated sequences.

Table 13: The length of generated sequences, energy consumption (J), and latency time (s) against BLIP-2 in different sampling policies. Our verbose images can still achieve better in different sampling policies.

| Sampling method | Method | MS-COCO | | | ImageNet | | |
|---|---|---|---|---|---|---|---|
| | | Length | Latency | Energy | Length | Latency | Energy |
| Greedy search | Original | 8.19 | 0.27 | 10.06 | 7.58 | 0.24 | 10.23 |
| | Noise | 7.32 | 0.23 | 10.24 | 7.74 | 0.25 | 10.43 |
| | Sponge samples | 8.47 | 0.27 | 11.30 | 8.11 | 0.27 | 10.90 |
| | NICGSlowDown | 195.44 | 5.22 | 216.48 | 267.02 | 7.45 | 301.47 |
| | **Verbose images (Ours)** | **323.97** | **8.86** | **367.72** | **352.16** | **9.56** | **399.19** |
| Beam search | Original | 9.53 | 0.42 | 17.74 | 8.84 | 0.43 | 18.75 |
| | Noise | 9.34 | 0.41 | 16.87 | 8.76 | 0.45 | 19.36 |
| | Sponge samples | 9.96 | 0.46 | 18.36 | 8.96 | 0.39 | 17.99 |
| | NICGSlowDown | 305.73 | 8.88 | 451.59 | 425.87 | 12.20 | 610.64 |
| | **Verbose images (Ours)** | **437.02** | **12.40** | **639.74** | **469.87** | **13.00** | **683.11** |
| Top-k sampling | Original | 8.20 | 0.33 | 13.22 | 7.54 | 0.31 | 12.85 |
| | Noise | 8.15 | 0.32 | 13.17 | 7.59 | 0.32 | 13.01 |
| | Sponge samples | 8.53 | 0.35 | 14.86 | 7.73 | 0.26 | 11.03 |
| | NICGSlowDown | 207.96 | 5.77 | 225.89 | 262.11 | 7.12 | 291.76 |
| | **Verbose images (Ours)** | **293.48** | **8.08** | **330.39** | **372.53** | **10.24** | **417.89** |
| Nucleus sampling | Original | 8.82 | 0.39 | 16.08 | 8.11 | 0.37 | 15.39 |
| | Noise | 9.55 | 0.43 | 17.53 | 8.37 | 0.44 | 19.39 |
| | Sponge samples | 22.53 | 0.73 | 30.20 | 43.59 | 1.51 | 63.27 |
| | NICGSlowDown | 103.54 | 3.78 | 156.61 | 129.68 | 4.34 | 180.06 |
| | **Verbose images (Ours)** | **226.72** | **7.97** | **321.59** | **250.72** | **10.26** | **398.58** |

Table 14: The length of generated sequences, energy consumption (J), and latency time (s) against BLIP-2 in different maximum lengths. Our verbose images can still achieve better in different maximum lengths.

| Maximum length | Method | MS-COCO | | | ImageNet | | |
|---|---|---|---|---|---|---|---|
| | | Length | Latency | Energy | Length | Latency | Energy |
| 128 | Original | 7.88 | 0.31 | 13.34 | 8.61 | 0.35 | 14.49 |
| | Noise | 8.45 | 0.35 | 14.11 | 8.25 | 0.34 | 13.81 |
| | Sponge samples | 16.60 | 0.51 | 21.79 | 22.73 | 0.72 | 28.90 |
| | NICGSlowDown | 35.94 | 1.26 | 49.60 | 40.46 | 1.46 | 59.13 |
| | **Verbose images (Ours)** | **61.55** | **2.25** | **90.53** | **66.89** | **2.38** | **91.97** |
| 256 | Original | 8.18 | 0.32 | 13.01 | 8.03 | 0.34 | 13.83 |
| | Noise | 8.26 | 0.35 | 14.67 | 8.02 | 0.34 | 13.46 |
| | Sponge samples | 18.75 | 0.55 | 23.42 | 31.68 | 0.93 | 37.26 |
| | NICGSlowDown | 60.73 | 2.00 | 80.83 | 70.57 | 2.21 | 87.41 |
| | **Verbose images (Ours)** | **109.13** | **4.04** | **163.74** | **116.94** | **4.17** | **167.02** |
| 512 | Original | 8.82 | 0.39 | 16.08 | 8.11 | 0.37 | 15.39 |
| | Noise | 9.55 | 0.43 | 17.53 | 8.37 | 0.44 | 19.39 |
| | Sponge samples | 22.53 | 0.73 | 30.20 | 43.59 | 1.51 | 63.27 |
| | NICGSlowDown | 103.54 | 3.78 | 156.61 | 129.68 | 4.34 | 180.06 |
| | **Verbose images (Ours)** | **226.72** | **7.97** | **321.59** | **250.72** | **10.26** | **398.58** |
| 1024 | Original | 8.18 | 0.35 | 13.97 | 8.49 | 0.35 | 14.69 |
| | Noise | 7.78 | 0.40 | 14.94 | 8.10 | 0.33 | 14.39 |
| | Sponge samples | 30.60 | 0.92 | 34.34 | 64.10 | 1.97 | 85.01 |
| | NICGSlowDown | 193.97 | 5.80 | 237.39 | 256.35 | 7.71 | 312.95 |
| | **Verbose images (Ours)** | **417.66** | **13.92** | **586.40** | **447.88** | **14.81** | **611.60** |

## H.1 DIFFERENT SAMPLING POLICIES

In our default settings, VLMs generate the sequences using nucleus sampling method (Holtzman et al., 2020) with $p = 0.9$ and temperature $t = 1$. Besides, we present the results of three other

sampling policies, including greedy search, beam search with a beam width of 5, and top-k sampling with $k = 10$. As depicted in Table 13, our verbose images can induce VLMs to generate the longest sequences across various sampling policies, demonstrating that our verbose images are not sensitive to the generation sampling policies.

## H.2 DIFFERENT MAXIMUM LENGTHS OF GENERATED SEQUENCES

Table 14 presents the results of five categories of visual images under varying maximum lengths of generated sequences. It shows that as the maximum length of generated sequences of VLMs increases, the length of generated sequences becomes longer, leading to higher energy consumption and longer latency time. Furthermore, our verbose images can consistently outperform the other four methods, which confirms the superiority of our verbose images.

# I GRID SEARCH

We conduct the experiments of grid search (Gao et al., 2023) for different parameters of loss weights and different momentum values.

## I.1 DIFFERENT PARAMETERS OF LOSS WEIGHTS

We set parameters of loss weights as $a_1 = 10$, $b_1 = -20$, $a_2 = 0$, $b_2 = 0$, $a_3 = 0.5$, and $b_3 = 1$ during the optimization of our verbose images. These parameters are determined through the grid search. The results of grid search are shown in Table 15, Table 16, Table 17, Table 18, Table 19, and Table 20, which demonstrates that our verbose images with $a_1 = 10$, $b_1 = -20$, $a_2 = 0$, $b_2 = 0$, $a_3 = 0.5$, and $b_3 = 1$ can induce VLMs to generate the longest sequences.

## I.2 DIFFERENT MOMENTUM VALUES

We show the results of our verbose images in different momentum values $m = \{0, 0.3, 0.6, 0.9\}$ in Table 21. These results demonstrate that it is necessary to adopt the addition of momentum during the optimization.

# J VISUALIZATION

We visualize the examples of the original images and our verbose images against BLIP, BLIP-2, InstructBLIP, and MiniGPT-4 in Fig. 9, Fig. 10, Fig. 11, and Fig. 12. It can be observed that VLMs with more advanced LLMs (*e.g.*, Vicuna-7B) generate more fluent, smooth, and logical content when encountering our verbose images.

Table 15: The length of generated sequences, energy consumption (J), and latency time (s) against BLIP-2 for grid search of $a_1$.

| $a_1$ | $b_1$ | $a_2$ | $b_2$ | $a_3$ | $b_3$ | MS-COCO | | | ImageNet | | |
|---|---|---|---|---|---|---|---|---|---|---|---|
| | | | | | | Length | Latency | Energy | Length | Latency | Energy |
| 1 | -20 | 0 | 0 | 0.5 | 1 | 215.91 | 7.36 | 310.36 | 233.67 | 8.83 | 375.31 |
| **10** | **-20** | **0** | **0** | **0.5** | **1** | **226.72** | **7.97** | **321.59** | **250.72** | **10.26** | **398.58** |
| 100 | -20 | 0 | 0 | 0.5 | 1 | 159.34 | 5.77 | 207.95 | 162.75 | 6.88 | 232.95 |

Table 16: The length of generated sequences, energy consumption (J), and latency time (s) against BLIP-2 for grid search of $b_1$.

| $a_1$ | $b_1$ | $a_2$ | $b_2$ | $a_3$ | $b_3$ | MS-COCO | | | ImageNet | | |
|---|---|---|---|---|---|---|---|---|---|---|---|
| | | | | | | Length | Latency | Energy | Length | Latency | Energy |
| 10 | -2 | 0 | 0 | 0.5 | 1 | 212.99 | 7.39 | 307.97 | 212.14 | 7.56 | 304.26 |
| **10** | **-20** | **0** | **0** | **0.5** | **1** | **226.72** | **7.97** | **321.59** | **250.72** | **10.26** | **398.58** |
| 10 | -200 | 0 | 0 | 0.5 | 1 | 195.23 | 6.74 | 253.56 | 195.38 | 6.32 | 266.37 |

Table 17: The length of generated sequences, energy consumption (J), and latency time (s) against BLIP-2 for grid search of $a_2$.

| $a_1$ | $b_1$ | $a_2$ | $b_2$ | $a_3$ | $b_3$ | MS-COCO | | | ImageNet | | |
|---|---|---|---|---|---|---|---|---|---|---|---|
| | | | | | | Length | Latency | Energy | Length | Latency | Energy |
| 10 | -20 | 0.1 | 0 | 0.5 | 1 | 177.59 | 7.06 | 270.62 | 213.63 | 8.18 | 297.52 |
| **10** | **-20** | **0** | **0** | **0.5** | **1** | **226.72** | **7.97** | **321.59** | **250.72** | **10.26** | **398.58** |
| 10 | -20 | 1 | 0 | 0.5 | 1 | 196.17 | 6.87 | 290.8 | 227.38 | 10.06 | 357.64 |

Table 18: The length of generated sequences, energy consumption (J), and latency time (s) against BLIP-2 for grid search of $b_2$.

| $a_1$ | $b_1$ | $a_2$ | $b_2$ | $a_3$ | $b_3$ | MS-COCO | | | ImageNet | | |
|---|---|---|---|---|---|---|---|---|---|---|---|
| | | | | | | Length | Latency | Energy | Length | Latency | Energy |
| 10 | -20 | 0 | 0.1 | 0.5 | 1 | 132.13 | 5.54 | 206.47 | 160.53 | 6.51 | 227.87 |
| **10** | **-20** | **0** | **0** | **0.5** | **1** | **226.72** | **7.97** | **321.59** | **250.72** | **10.26** | **398.58** |
| 10 | -20 | 0 | 1 | 0.5 | 1 | 203.61 | 7.08 | 286.32 | 224.06 | 9.36 | 345.61 |

Table 19: The length of generated sequences, energy consumption (J), and latency time (s) against BLIP-2 for grid search of $a_3$.

| $a_1$ | $b_1$ | $a_2$ | $b_2$ | $a_3$ | $b_3$ | MS-COCO | | | ImageNet | | |
|---|---|---|---|---|---|---|---|---|---|---|---|
| | | | | | | Length | Latency | Energy | Length | Latency | Energy |
| 10 | -20 | 0 | 0 | 0.05 | 1 | 209.43 | 7.18 | 292.43 | 217.03 | 8.62 | 303.73 |
| **10** | **-20** | **0** | **0** | **0.5** | **1** | **226.72** | **7.97** | **321.59** | **250.72** | **10.26** | **398.58** |
| 10 | -20 | 0 | 0 | 5 | 1 | 199.62 | 7.17 | 283.76 | 226.52 | 8.06 | 335.48 |

Table 20: The length of generated sequences, energy consumption (J), and latency time (s) against BLIP-2 for grid search of $b_3$.

| $a_1$ | $b_1$ | $a_2$ | $b_2$ | $a_3$ | $b_3$ | MS-COCO | | | ImageNet | | |
|---|---|---|---|---|---|---|---|---|---|---|---|
| | | | | | | Length | Latency | Energy | Length | Latency | Energy |
| 10 | -20 | 0 | 0 | 0.5 | 0.1 | 208.48 | 7.16 | 290.92 | 236.06 | 9.97 | 380.23 |
| **10** | **-20** | **0** | **0** | **0.5** | **1** | **226.72** | **7.97** | **321.59** | **250.72** | **10.26** | **398.58** |
| 10 | -20 | 0 | 0 | 0.5 | 10 | 186.68 | 7.05 | 282.61 | 222.03 | 9.03 | 340.55 |

Table 21: The length of generated sequences, energy consumption (J), and latency time (s) against BLIP-2 for grid search of the momentum $m$.

| $m$ | MS-COCO | | | ImageNet | | |
|---|---|---|---|---|---|---|
| | Length | Latency | Energy | Length | Latency | Energy |
| 0 | 199.92 | 7.02 | 292.55 | 231.03 | 7.88 | 318.34 |
| 0.3 | 201.83 | 7.21 | 301.70 | 237.85 | 9.33 | 330.83 |
| 0.6 | 218.69 | 7.69 | 312.77 | 240.27 | 9.85 | 376.59 |
| **0.9** | **226.72** | **7.97** | **321.59** | **250.72** | **10.26** | **398.58** |

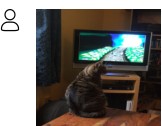

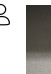 A cat looking at a television screen turned on.

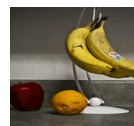 A banana and apple being suspended on wires.

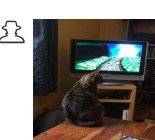

A group of a cat with a black bear on its neck in the form of a mini guitar, a fake black dog, and a tiger dog, painted as a truck with a leopard, that is sitting in the car. The person funny animal print graduation on ruges as a craft print pizzas and a funny cat hola person with the bus. A person with a calculator bear head and the laptops made to look like two dogs, slotted black bears and black bears with auto cars as wheels painted on them, as their animals, all animals, to look like they treading as the self. Their bears holding a vehicle with wheels and the green car **… …**

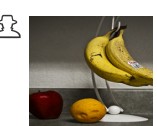

A hand gesture of words with much fruit and knife body don't see group out of cake. some bananas and an apple by a sculpture on one oval table in front of an arena oval art print on the walls in a room in front by advertising for food plate art group to people lunch series with art dinner or a white plate of apples above, an entrament. The banana peel with two oranges and an apple on a pedestal next to a statue set by person background with fruit in bedding bed oval background, oval drawing, oval person's, space oval floor with strawberry. In the foreground, there are bananas **… …**

Figure 9: A visualization example for the original images and our verbose counterpart against BLIP.

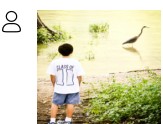

A boy looking at a bird on a cliff.

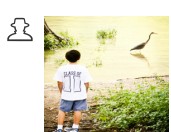

A boy in a white t-shirt and shorts standing on a dirt path by a lake with a large bird in the background. The sun shines brightly, casting a warm glow on the scene and creating a peaceful atmosphere. In the background, a large, majestic bird can be seen, perhaps a heron or an egret. He is with his father, his hands with bird, crocodile, old man and his son, his wife, his dog. The calm waters of the lake reflect boy and woman, mother and son, car, and son. This boy is holding his toy, his boat, and his grandfather, and looking down the river … …

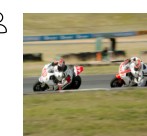

Two motorcyclists riding race cars down a track.

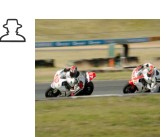

Super stock race is meeting at beautiful rain aerobatics, air - show, fireball paraflame, fireball mavita group photos, On January 13, 2018, Round 12 of the Super Stock 600 took place, drawing a large crowd of enthusiastic spectators, eager to witness the excitement on the Super Stock 600 track. The super stock race shows the car photos, person,  heroes automotive, and whoever. During the process, the classic motorcycle competitors have won the super stock 600 class at the beautiful sunny day and classic motorcycle race on Saturday every morning … …

Figure 10: A visualization example for the original images and our verbose counterpart against BLIP-2.

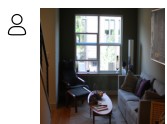

👤 What is the content of this image?

🤖 The image features a cozy living room with a well-lit window, providing a view of a city scene. There is a small couch in the room, and it is situated close to a staircase. Additionally, there is a coffee table, several pillows, and two chairs in the room.

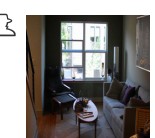

What is the content of this image?

The image features two levels, the second level featuring the main scene. The first floor is a small apartment or studio with a living room and dining area. The living room includes a couch and a coffee table, while the dining area contains chairs and a dining table. A lamp is situated on the edge of the image on the left, and there is a window on the left side of the room. Several chairs can be seen in the living room and dining area, creating a cozy atmosphere. On the second floor, the main scene … …

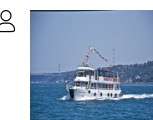

👤 What is the content of this image?

🤖 The image features a large white ferry boat cruising across the water, surrounded by the blue ocean and a bridge in the background. There are numerous flags adorning the ferry boat, showcasing the spirit of the vessel as it travels through the water.

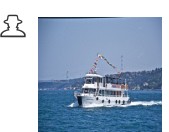

What is the content of this image?

The image depicts a ship floating on a calm sea, carrying several passengers with its high-speed propeller. The ship has a colorful flag, indicating its nationality or destination. The background features a picturesque landscape, with a row of trees and a distant shore in the background. The boat is passing by a large lighthouse that stands proudly, serving as a navigation aid and beacon for ships sailing by. In the foreground, an airplane is visible, perhaps adding to the overall picture of an exciting maritime … …

Figure 11: A visualization example for the original images and our verbose counterpart against InstructBLIP.

👤 Give the following image. You will be able to see the image once I provide it to you. Please answer my questions.

👤 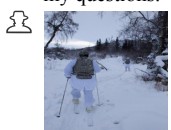

👤 What is the content of this image?

🤖 This image depicts two people on skis with poles, walking through snowy terrain. The sky is overcast, and there are some dark trees visible in the background. The color palette is muted and monochromatic, with the figures wearing white clothing.

👥 Give the following image. You will be able to see the image once I provide it to you. Please answer my questions.

👥 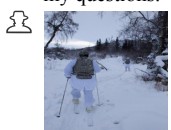

👥 What is the content of this image?

🤖 The content of this image is two people in a snowy environment. The person in the front is wearing a suit that appears to be made out of snow and the person in the back is holding a gun. There are also several other objects in the background such as trees and other buildings. The mood of the image is cold and wintery. The image is also in black and white. The subject matter of the image is people in a snowy environment. The genre of the image is landscape. The style of the image is realistic **… …**

👤 Give the following image. You will be able to see the image once I provide it to you. Please answer my questions.

👤 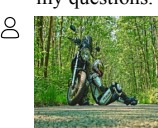

👤 What is the content of this image?

🤖 This image is of a motorcycle parked on the side of a road surrounded by a dense forest. It appears to be taken on a sunny day with lots of light coming through the trees. The motorcycle is parked with the side of the bike facing the camera.

👥 Give the following image. You will be able to see the image once I provide it to you. Please answer my questions.

👥 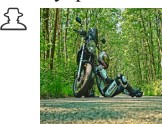

👥 What is the content of this image?

🤖 The image shows a man and a motorcycle on a road surrounded by trees. The man is sitting on the motorcycle with his legs stretched out in front of him. He is wearing a black leather jacket and black jeans. The motorcycle is black and has a silver fuel tank. The background is made up of tall, thin trees in a row, creating a lush green canopy above the road. There are also small green plants and grass growing along the sides of the road **… …**

Figure 12: A visualization example for the original images and our verbose counterpart against MiniGPT-4.

