# OpenReview forum: "Inducing High Energy-Latency of Large Vision-Language Models with Verbose Images"
_ICLR.cc/2024/Conference — ICLR 2024 poster_

### Official Review · Reviewer_NtLX · 2023-10-28

**Soundness:** 2 fair
**Presentation:** 3 good
**Contribution:** 3 good
**Rating:** 6
**Confidence:** 4

**Summary:**

The paper investigates the attack of VLMs, which will induce high energy-latency cost during inference via maximizing the length of the sequence. To achieve this aim, the paper proposes three losses, which first decrease the probability of the end-of-sequence token, and then increase the uncertainty of output tokens and diversity of token embeddings. Experiments are done on popular VLMs models including BLIP, BLIP-2, InstructBLIP, and MiniGPT-4, and on common datasets including MS-COCO and ImageNet datasets.

**Strengths:**

* The paper investigates an important and interesting problem, which attacks the inference efficiency of VLMs. The proposed solution is simple and does not cost much.
* The paper is well-written and easy to follow. The figures are presented well.
* Experiments have been done on a variety of models with two tasks. Improvements in sequence length and latency are obvious. Also, the paper gives a detailed ablation study.

**Weaknesses:**

* The motivation or the reason for the effectiveness of the second loss is weak. I think the second loss, which increases the uncertainty of tokens is more like destroying the accurate outputs. While the paper claims that the second loss can help with longer sequences, I do not see a strong correlation. Can the authors give some description to build the relationship between sequence length and token uncertainty, which usually indicates less favorable accuracy performance?

* In the third loss part, the paper claims that “for a matrix, the calculation of the matrix rank is an NP-hard non-convex problem”, and thus proposes the third loss. Unfortunately, this is not true. For the 2-way matrix that corresponds to [token, hidden_states] in this example, the rank calculation is available by Gaussian elimination and is not NP-hard. Therefore, I keep doubting the motivation and design of the third loss, which also shows the least performance improvement in the ablation study.

* The paper seems to target the problem in VLMs and claims that VLMs have special things like vulnerabilities from the manipulation of visual inputs. While it says that other methods for LLMs, can not be directly applied to VLMs, I do not see the special design or consideration in this method for challenges in VLMs.

**Questions:**

Please check the weakness part.

---

> ### Author Response · Authors · 2023-11-20
> **Response to Reviewer NtLX (Part I)**
>
> **Q1: The motivation or the reason for the effectiveness of the second loss is weak. I think the second loss, which increases the uncertainty of tokens is more like destroying the accurate outputs. While the paper claims that the second loss can help with longer sequences, I do not see a strong correlation. Can the authors give some description to build the relationship between sequence length and token uncertainty, which usually indicates less favorable accuracy performance?**
> **A:** Thank you for your comment regarding the motivation and effectiveness of the second loss in our study. We will try our best to introduce the motivation of our proposed second loss.
> To generate a longer sequence, we propose to induce the output to deviate from the original sequence. This is because with the guide of the Delayed EOS loss (the first loss), the generated sequence can be deviated towards the direction of a longer sequence away from the original auto-gressive dependency. Then, **the problem is formulated as how to induce the generated sequence away from the original sequence.**
> To provide clearer expression, we first revisit the mathematical preliminaries of VLMs as follows. Given an image $x$, an input text $c_{\text{in}}$ and a sequence of generated output tokens $y=[y_1, y_2, ..., y_N]$, where $y_i$ represents the $i$-th generated token, $N$ is the length of the output sequence, VLMs generally generate each token with the highest probability. These tokens form a generated sequence of VLMs. Moreover, the probability of the generated sequence can be calculated as a product of conditional probabilities of each token [1].$$P(y_1...y_N;x;c_{\text{in}})=\prod \limits_{i=1}^N P(y_i|y_1,...,y_{i-1};x;c_{\text{in}})$$Consequently, the probability of the generated sequence is also the highest probability. **To deviate the original generated sequence, we can reduce the probability of the generated sequence, which can be achieved by lowering the probability of each token.** To accomplish this, we propose to increase the entropy of each token, which in turn increases the token uncertainty.
>
>
> [1] Bengio Y, Ducharme R, Vincent P. A neural probabilistic language model. In Advances in neural information processing systems (NeurIPS), 2000, 13.
>
> **Q2: In the third loss part, the paper claims that “for a matrix, the calculation of the matrix rank is an NP-hard non-convex problem”, and thus proposes the third loss. Unfortunately, this is not true. For the 2-way matrix that corresponds to [token, hidden_states] in this example, the rank calculation is available by Gaussian elimination and is not NP-hard. Therefore, I keep doubting the motivation and design of the third loss, which also shows the least performance improvement in the ablation study.**
> **A:** Thanks for your insightful comment!
> - We apologize for the unclarify in our claim regarding the third loss and appreciate your valuable feedback which helps us to identify this issue. In our original paper, we stated, "However, for a matrix, the calculation of the matrix rank is an NP-hard non-convex problem, and we can not directly optimize the rank of a matrix." Our intention was to convey that "In general, the optimization of Rank Minimization Problem (RMP) is known to be computationally intractable (NP-hard)", as mentioned on the second page of Chapter 1 in the referenced thesis paper [1]. To provide a clearer expression, we have corrected our paper accordingly.
> - The motivation of the third loss is as follows. The primary objective of both the second and third losses is to deviate the output from the original sequence, effectively breaking the original dependency. The second loss is designed at the token level. By enhancing the entropy of each token, we aim to lower the probability of the whole sequence, ultimately leading to a sentence that deviates from the original dependency, as detailed in *[Q1 in Response to Reviewer NtLX].* In contrast, **the third loss is designed at the sequence level, focusing on directly destroying the relationship among tokens in a sequence.** To represent the overall relationship among tokens in a sequence, we argue that the rank of the concatenated matrix of hidden states among all generated tokens can be a suitable choice. Hence, combined with the estimition of rank optimization [1], we propose the third loss to improve token diversity, which can induce VLMs to generate longer sequence. Based on the ablation study for three losses, we can observe that the third loss contributes to performance improvement.
>
> [1] Fazel M. Matrix rank minimization with applications. PhD thesis, Stanford University, 2002.

---

> ### Author Response · Authors · 2023-11-20
> **Response to Reviewer NtLX (Part II)**
>
> **Q3: The paper seems to target the problem in VLMs and claims that VLMs have special things like vulnerabilities from the manipulation of visual inputs. While it says that other methods for LLMs, can not be directly applied to VLMs, I do not see the special design or consideration in this method for challenges in VLMs.**
> **A:** Thanks for your valuable advice! The challenges in VLMs arise primarily from the visual input manipulation, both visual and textual manipulation, and sampling policies.
> - **Visual input manipulation.** Compared with LLMs, VLMs incorporate the vision modality into impressive LLMs to enable visual process ability. Our verbose images focus on manipulating the visual input to exploit the vulnerabilities introduced by the integration of vision, following [1,2].  In intuition, different textual prompts can lead to a larger difference in the length of generated sequences than different input images, such as "What is the content of this image?" and "Describe the image as detailed as possible." We first argue that **a substantial difference in the length of generated sequences can also be achieved by using two almost similar images**, which only have a slight difference in perturbations.
> - **Both visual and textual manipulation.** VLMs are capable of processing both visual and textual inputs, enabling them to handle multi-modal tasks effectively. As shown in Table 7 of our original paper, our verbose images can adapt to different textual prompts and exhibit superior performance compared to other methods. Besides, we also optimize both visual and textual inputs to induce high energy-latency cost. For the optimization of textual inputs, we adopt a parameterized distribution matrix to optimize input textual tokens, as suggested in [3]. Table 8 of our original paper also demonstrate the effectiveness of our proposed method.
> - **Sampling policies.** Current VLMs generate sequences based on the advanced sampling policies with randomness. Hence, VLMs generate different sequences for the same input data. This variability in specific generated tokens makes it challenging to optimize the loss based on specific generated tokens. Different from the objective in [4], we acknowledge this challenge and propose three losses **that all don't involve the specific tokens** for our verbose images. These losses are designed to effectively exploit the vulnerabilities of VLMs while addressing the inherent randomness in the generated sequences. Our experimental results validate that our verbose images can induce higher energy-latency costs in VLMs, demonstrating the effectiveness of our verbose images in tackling the unique challenges associated with VLMs and their sampling policies.
>
> [1] Bagdasaryan E, Hsieh T Y, Nassi B, et al. (Ab) using Images and Sounds for Indirect Instruction Injection in Multi-Modal LLMs. In 2023.
> [2] Qi X, Huang K, Panda A, et al. Visual Adversarial Examples Jailbreak Large Language Models. In 2023.
> [3] Guo C, Sablayrolles A, Jégou H, et al. Gradient-based adversarial attacks against text transformers. In Annual Meeting of the Association for Computational Linguistics (ACL). 2021.
> [4] Chen S, Song Z, Haque M, et al. Nicgslowdown: Evaluating the efficiency robustness of neural image caption generation models. In Proceedings of the IEEE/CVF Conference on Computer Vision and Pattern Recognition (CVPR). 2022: 15365-15374.

---

> ### Comment · Reviewer_NtLX · 2023-11-20
> **post-rebuttal**
>
> Thank you for your detailed reply. Most of my concerns have been addressed. I'd like to increase my score to 6.

---

> ### Author Response · Authors · 2023-11-22
> **Thanks for Reviewer NtLX's Response**
>
> Thank you for acknowledging our clarifications and for your willingness to update the score. Your valuable feedback are instrumental in improving the quality of our work.

---

### Official Review · Reviewer_Ysyk · 2023-11-01

**Soundness:** 3 good
**Presentation:** 3 good
**Contribution:** 2 fair
**Rating:** 6
**Confidence:** 4

**Summary:**

This paper propose an attack that makes a VLM model generate a very long sequence such that its power consumption become much higher. The paper adopts existed approaches and augments them with two new design into the training framework.

**Strengths:**

1. The proposed method demonstrates the attack power on VLM and make the model generate very long output.

2. The paper proposed new loss ($L_3$) and temporal weighting to train the model better.

**Weaknesses:**

1. A simple defense of the service provider could be set the limit of token based on the statistics of the clean data. As shown in Fig. 5 and 6, the original distribution focus certain length. In such case, it seems that this attack won't be too sever anymore?

2. I am not sure is this a real problem or not, as pointed by 1, a simple approach can resolve it (but with some tradeoff); on the other hand, as the attacked results do not make sense anymore (e.g. Fig.8), it seems the attacks can be easy identified. I think the problem definition should be constrained on the results are still make sense and good. That is, the evaluation should consider the performance of those tasks.

**Questions:**

1. Why the quality output won't be a metric? Does it mean that this attack are designed to be easily spotted? As one of current practice is using another LLM to inspect the output of current LLM.

2. In Fig. 3, 5 and 6, why the attack distribution results in like two-mode distribution?

3. What is the image embedding distance between the original image and the attacked image?

4. What is the energy consumption for generating one attack image?

5. When showing the results of length in the Table, the standard deviation should be added.

---

> ### Author Response · Authors · 2023-11-20
> **Response to Reviewer Ysyk (Part I)**
>
> **Q1: A simple defense of the service provider could be set the limit of token based on the statistics of the clean data. As shown in Fig. 5 and 6, the original distribution focus certain length. In such case, it seems that this attack won't be too sever anymore?**
> **A:** Thank you for your valuable feedback and for pointing out the potential defense strategy. We have added a new discussion about a simple defense strategy that limits the generation length in Appendix G.1 in our revised paper. We would like to point out the infeasibility from the following perspectives：
> - We argue that it is **difficult to compute the exact statistics of the clean data** because service providers can not have access to user-specific input data. Users have diverse requirements and input data, leading to a wide range of sentence lengths and complexities. For example, the prompt text of "Describe the given image in one sentence." and "Describe the given image in details." can introduce different lengths of generated sentences. We visualize **a case in Fig. 8 of our revised paper.** Consequently, service providers often consider a large token limit to accommodate these diverse requirements and ensure that the generated sentences are complete and meet users' expectations. Previous work [1] also states a same view as us, specifically in Section 3 in [1]. In Fig. 5 and Fig. 6, the length distribution is shown with the fixed prompt text to ensure the consistency of our experiments. We also show the results with different prompt texts in Table 7 of our original paper. All these results can verify that our verbose images are **adaptable for different prompt texts** and can induce the length of generated sentences closer to the token limit set by the service provider. As a result, the energy-latency cost can be increased while staying within the imposed constraints.
> - We argue that this attack surface about availability of VLMs becomes more important and our verbose images can **induce more serious attack consequences in the era of large (vision) language models.** The development of VLMs and LLMs has led to models capable of generating longer sentences with logic and coherence. Consequently, service providers have been increasing the maximum allowed length of generated sequences to ensure high-quality user experiences. For instance, gpt-3.5-turbo and gpt-4-turbo allow up to **4,096** and **8,192** tokens, respectively. Hence, we would like to uncover that while longer generated sequences can indeed improve service quality, they also introduce potential security risks about energy-latency cost, as our verbose images demonstrates. Therefore, when VLM service providers consider increasing the maximum length of generated sequences for better user experience, they should **not only focus on the ability of VLMs but also take the maximum energy consumption payload into account.**
>
>
> [1] Chen S, Song Z, Haque M, et al. Nicgslowdown: Evaluating the efficiency robustness of neural image caption generation models. In Proceedings of the IEEE/CVF Conference on Computer Vision and Pattern Recognition (CVPR). 2022: 15365-15374.

---

> ### Author Response · Authors · 2023-11-20
> **Response to Reviewer Ysyk (Part II)**
>
> **Q2-1: I am not sure is this a real problem or not, as pointed by 1, a simple approach can resolve it (but with some tradeoff)**
> **A:** Thanks for your insightful comments!
> - We argue that this is a real problem and this attack can induce more serious attack consequences in the era of large (vision) language models. The maximum generated length is increasing for the better response but also introduces the risk of the potential energy-latency collapse. Hence, it is necessary to evaluate the worst-case energy-latency cost during the deployment of VLMs. More details are in *[Q1 in Response to Reviewer Ysyk].*
> - Besides, previous study [1,2,3] has explored this attack in smaller-scale models and LLMs. Hence, we think the vulnerability about the energy-latency cost should also be noticed further for VLMs and we propose our verbose images to induce high energy-latency cost first tailored for VLMs. The experimental results show that our verbose images can generate longer sentences than previous studies on four open-sourced VLMs for reproducibility.
>
> [1] Chen S, Song Z, Haque M, et al. Nicgslowdown: Evaluating the efficiency robustness of neural image caption generation models. In Proceedings of the IEEE/CVF Conference on Computer Vision and Pattern Recognition (CVPR). 2022: 15365-15374.
> [2] Shumailov I, Zhao Y, Bates D, et al. Sponge examples: Energy-latency attacks on neural network. In IEEE European symposium on security and privacy (EuroS&P). IEEE, 2021: 212-231.
> [3] Chen S, Liu C, Haque M, et al. Nmtsloth: understanding and testing efficiency degradation of neural machine translation systems. In Proceedings of the 30th ACM Joint European Software Engineering Conference and Symposium on the Foundations of Software Engineering (ESEC/FSE). 2022: 1148-1160.
>
>
> **Q2-2: On the other hand, as the attacked results do not make sense anymore (e.g. Fig.8), it seems the attacks can be easy identified. I think the problem definition should be constrained on the results are still make sense and good. That is, the evaluation should consider the performance of those tasks.**
> **A:** Thanks for your insightful comment! We have added a new discussion about the model performance of three energy-latency attacks in Appendix G.2 in our revised paper.
> - As you suggested, we evaluate the BLEU and CIDEr scores for sponge samples, NICGSlowDown, and our verbose images on MS-COCO for BLIP-2, as illustrated in the below table. Concretely, our verbose images generate the longest sequence, extending it to 226.72, while sponge samples, despite their superior captioning performance, only increase the length to 22.53. Furthermore, both the length of the generated sequence and the model performance of our verbose images outperform those of NICGSlowDown.
> - We argue that although the performance degradation can be detected, it is crucial to note that **the attack has already been executed before the performance is evaluated.** In other words, the energy-latency cost has been first increased and then this attack is detected.
> - Besides, similar with the observation in Section 5.4.1 in [1], we observe that our verbose images not only induces high energy-latency cost of VLMs but also achieves **an untargeted attack,** which uncovers more security vulnerabilities from VLMs. Notably, your advice that the problem definition should be constrained to render results still good is very insightful and we will consider it in our future work.
>
> |         |  Length | Latency | Energy | BLEU-1 | BLEU-2 | BLEU-3 | BLEU-4 | CIDEr |
> |:-------:|:-------:|:-------:|:------:|:------:|:------:|:------:|:------:|:-----:|
> |  Sponge samples |  22.53  |   0.73  |  30.20 |  0.298 |  0.193 |  0.120 |  0.073 | 0.571 |
> |   NICGSlowDown  | 103.54  |   3.78  | 156.61 |  0.012 |  0.005 |  0.002 |  0.001 | 0.027 |
> | Verbose images (Ours) |  226.72 |   7.97  | 321.59 |  0.026 |  0.012 |  0.005 |  0.003 | 0.099 |
>
> [1] Chen S, Song Z, Haque M, et al. Nicgslowdown: Evaluating the efficiency robustness of neural image caption generation models. In Proceedings of the IEEE/CVF Conference on Computer Vision and Pattern Recognition (CVPR). 2022: 15365-15374.

---

> ### Author Response · Authors · 2023-11-20
> **Response to Reviewer Ysyk (Part III)**
>
> **Q3: Why the quality output won't be a metric? Does it mean that this attack are designed to be easily spotted? As one of current practice is using another LLM to inspect the output of current LLM.**
> **A:** Thanks for your insightful suggestion!
> - We have listed the BLEU and CIDEr scores for sponge samples, NICGSlowDown, and our verbose images on MS-COCO for BLIP-2, as shown in the table in *[Q2-2 in Response to Reviewer Ysyk]*. We acknowledge that this attack can be detected by the output performance. However, we argue that it is important to note that **the attacks have already been executed** before the performance or output performance is evaluated.
> - Thank you for your suggestion to use another LLM to inspect the output of the current LLM as a potential defense mechanism against our verbose images. We acknowledge that this approach can be effective in detecting and mitigating the impact of attacks on the system. However, it is important to consider the cost implications of implementing such a solution. Utilizing a second LLM to inspect the output of the first LLM would **double the service cost,** as both models would need to be maintained, updated, and monitored.
>
> **Q4: In Fig. 3, 5 and 6, why the attack distribution results in like two-mode distribution?**
> **A:** Thank you for your insightful comment. We have added the explanation to clarify why the length distribution of our verbose images results in bimodal distribution (two-mode distribution) in Appendix D in our revised paper. We agree that the length distribution of generated sequences in the four VLM models exhibits a two-mode distribution (bimodal distribution). Specifically, our verbose images tend to prompt VLMs to generate either long or short sequences. We conjecture the reasons as follows. A majority of the long sequences are generated, confirming the effectiveness of our verbose images. As for the short sequences, we have carefully examined the generated content and observed two main cases. **In the first case,** our verbose images fail to induce long sentences, particularly for BLIP and BLIP-2, which lack instruction tuning and have smaller parameters. **In the second case,** our verbose images can confuse the VLMs, leading them to generate statements such as "I'm sorry, but I cannot describe the image." This scenario predominantly occurs with InstructBLIP and MiniGPT-4, both of which have instruction tuning and larger parameters.
>
>
> **Q5: What is the image embedding distance between the original image and the attacked image?**
> **A:** Thanks for your valuable comment. We have added experimental results of image embedding distance between the original image and the verbose image in Appendix G.3 in our revised paper. We first adopt the image encoder of CLIP to extract the image embedding. Then the image embedding distance is calculated as the cosine similarity of both original images and the corresponding sponge samples, NICGSlowDown, and our verbose images. The results are shown in below tables which demonstrate that these methods are **similar in image embedding distance.**
>
> The results on MS-COCO is as follows.
> |                |  BLIP | BLIP-2 | InstructBLIP | MiniGPT-4 |
> |:--------------:|:-----:|:------:|:------------:|:---------:|
> | Sponge samples | 0.965 |  0.977 |     0.978    |   0.978   |
> |  NICGSlowDown  | 0.966 |  0.978 |     0.979    |   0.971   |
> | Verbose images (Ours) | 0.965 |  0.970 |     0.971    |   0.969   |
>
> The results on ImageNet is as follows.
> |                |  BLIP | BLIP-2 | InstructBLIP | MiniGPT-4 |
> |:--------------:|:-----:|:------:|:------------:|:---------:|
> | Sponge samples | 0.968 |  0.979 |     0.981    |   0.981   |
> |  NICGSlowDown  | 0.968 |  0.980 |     0.981    |   0.975   |
> | Verbose images (Ours) | 0.966 |  0.969 |     0.970    |   0.971   |

---

> ### Author Response · Authors · 2023-11-20
> **Response to Reviewer Ysyk (Part IV)**
>
> **Q6: What is the energy consumption for generating one attack image?**
> **A:** Thanks for your valuable comment! We have added experimental results of energy consumption for generating one verbose image in Appendix G.4 in our revised paper.
> - We calculate the energy-latency cost for the generation of one verbose image and show the results in the below table. It can be observed that the energy-latency cost during attack increases with the number of attack iterations, along with an increase in energy consumption for generating one verbose image.
> - Besides, the overall energy-latency cost of generating a single verbose image is higher than that of using it to attack VLMs. Therefore, it is necessary for the attacker to make full use of every generated verbose image and learn from DDoS attack strategies to perform this attack more effectively. Specifically, **the attacker can instantly send as many copies of the same verbose image as possible to VLMs,** which increases the probability of exhausting the computational resources and reducing the availability of VLMs service. Once the attack is successful and causes the competitor's service to collapse, the attacker will acquire numerous users from the competitor and gain significant benefits.
>
> | Attack iterations | Length during attack | Latency during attack | Energy during attack | Latency during generation | Energy during generation |
> |:-----------------:|:--------------------:|:---------------------:|:--------------------:|:-------------------------:|:------------------------:|
> |         0         |         8.40         |         0.45          |         17.91        |             0             |             0            |
> |        100        |        70.26         |         3.77          |        136.73        |           50.28 |	6593.69        |
> |        500        |        175.24        |         6.84          |        285.63        |          261.20 |	36537.34|
> |        1000       |        226.72        |         7.97          |        321.59        |           442.12 |	67230.84 |
>
> **Q7: When showing the results of length in the Table, the standard deviation should be added.**
> **A:** Thanks for your constructive comment! We have supplemented the standard deviation results of our main table  in Appendix G.5 in our revised paper, and due to time constraints, we will update other experiments in our revised versions.

---

> > ### Comment · Reviewer_Ysyk · 2023-11-22
> > **Thanks for authors' response**
> >
> > Thanks for the response, it addressed my most of questions, and I raise my score to 6 since this is an interesting research and authors did a comprehensive study. Nonetheless, I still feel the cost to generate an attack is way too expensive given the power consumption reported in the rebuttal. For the best case, it costs about 50 times power; that means, if the defender will to spend 2-3 times effort to invalid the generated image, the attacker actually will cost more than the defender.

---

> ### Author Response · Authors · 2023-11-22
> **Thanks for Reviewer Ysyk's Response**
>
> Thank you for acknowledging our clarifications and for your willingness to update the score. Your insightful suggestions are very helpful in improving the quality of our work.

---

### Official Review · Reviewer_xu4N · 2023-11-01

**Soundness:** 4 excellent
**Presentation:** 4 excellent
**Contribution:** 3 good
**Rating:** 8
**Confidence:** 3

**Summary:**

The computation cost, including energy consumption and latency time, is an important issue for the development of large vision-language models. This paper investigates the computation cost and observes that it highly correlates with the generated sequence length. To maximize the generated sequence length, this paper seeks to craft an imperceptible perturbation during the inference stage and proposes three losses to guide the derivation of the verbose image, including a delayed EOS loss, an uncertainty loss and a token diversity loss. In addition, a temporal weight adjustment algorithm is proposed for better optimization. Experiments on MSCOCO and ImageNet show the effectiveness of the proposed verbose images. Beyond inducing longer generated sequences, the verbose images can introduce more dispersed attention and enhance the object hallucination.

**Strengths:**

1. The problem is well-formulated. The solutions and results are delivered clearly. I appreciate the writing and presentation.
2. Simple yet effective methods to induce longer generated sequences. Adding imperceptible perturbation is a reasonable way to induce longer sequences. Optimizing the verbose images via the sequence losses is also quite straightforward.
3. The experimental results are quite convincing. The models, when faced with verbose images, perform much worse than original results. The ablations and qualitative explorations show intriguing results.

**Weaknesses:**

Despite the simplicity and effectiveness, I think this paper lacks novelty.

**Questions:**

The verbose images reveal some susceptible aspects of current LVLMs. I am interested to see how such findings can further shed light on the learning of LVLMs. By the way, I suspect whether such a phenomenon always exists in the data-driven deep-learning approach? So, is there any insight on optimizing LVLMs?

---

> ### Author Response · Authors · 2023-11-20
> **Response to Reviewer xu4N (Part I)**
>
> **Q1: Despite the simplicity and effectiveness, I think this paper lacks novelty.**
> **A:** We appreciate reviewer's positive feedback on the simplicity and effectiveness of our method. We would like to further clarify our novelty and contributions from the following aspects:
> - **As the first work,** we investigate the energy-latency cost manipulation in VLMs, which demonstrates that such manipulation can potentially lead to a severe collapse of service. We observe that the energy-latency cost exhibits an approximately positive linear relationship with the length of generated sequences in VLMs. Based on this observation, the problem of inducing high energy-latency cost can be formulated as increasing the length of generated sequences.
> - **We propose verbose images** to craft an imperceptible perturbation to induce high energy-latency cost for VLMs. The Delayed EOS Loss is proposed to encourage VLMs to generate longer sequences by delaying the appearance of the EOS token. Moreover,  Uncertainty Loss and Token Diversity Loss are proposed to deviate the generated sentence from the original sentence in both token-leval and sequence-level. The motivation behind these two losses is described in more detail in *[Q1 and Q2 in Response to Reviewer NtLX]*. What is more important, we present the first effective solution to obtain **an effective attack to induce high energy-latency cost of VLMs** (i.e. increasing the length of generated sequences by delaying EOS token and breaking original dependency in both token-leval and sequence-level), and will inspire more researchs to explore more advanced methods along this direction.
> - **Besides, we also provide interpretation from the experimental results for VLMs.** Our experimental results show that our verbose images can successfully induce VLMs to generate a longer sequence, increasing the length of generated sequences by 7.87× and 8.56× compared to original images on MS-COCO and ImageNet datasets. In terms of visual interpretation, our verbose images prompt VLMs to exhibit more dispersed attention across the entire image. For the textual interpretation, the generated longer sequences from our verbose images can include additional objects that are not present in the image. We argue that our verbose images can provide valuable insights into the behavior of large vision-language models (VLMs). More details on insights please refer to *[Q2-1 in Response to Reviewer xu4N].*
>
> In summary, given the status that there is no effective energy-latency cost manipulation in VLMs, we not only propose the first effective soution, but also provide interpretaion from experimental results. We believe this work is valuable and will be of great interest to the community of foundation model security.

---

> ### Author Response · Authors · 2023-11-20
> **Response to Reviewer xu4N (Part II)**
>
> **Q2-1: The verbose images reveal some susceptible aspects of current LVLMs. I am interested to see how such findings can further shed light on the learning of LVLMs.  So, is there any insight on optimizing LVLMs?**
> **A:** Thanks for your great question which points out the potential border impact of our work! Our experiments show that our verbose images can not only induce high energy-latency cost but also can generate sequences with hallucinated objects, revealing the susceptibility of current VLMs to hallucination. We argue that our proposed verbose images provide **a new direction for investigating the underlying mechanisms of hallucination generation.** By utilizing verbose images as new tools, we can further explore and understand the reasons behind hallucination generation. We will consider these aspects as future work, building upon the insights gained from our current research. We appreciate your valuable comments and insights again, which really helps to boost our thinking and shape the direction of our future work on verbose images.
>
>
> **Q2-2: By the way, I suspect whether such a phenomenon always exists in the data-driven deep-learning approach?**
> **A:** Thanks for your insightful comment! The energy-latency vulnerability has been explored in the image classification models [1], image captioning models [2], LLMs for machine translation systems [1,3], multi-exit neural network [4,5], and the lidar-based detection [6]. Hence, such a phenomenon most exists in the data-driven deep learning approach. We argue that energy-latency vulnerability is more critical for recent advanced large-scale generation models, such as VLMs, because they necessitate substantial computational resources for deployment.
>
> [1] Shumailov I, Zhao Y, Bates D, et al. Sponge examples: Energy-latency attacks on neural network. In IEEE European symposium on security and privacy (EuroS&P). IEEE, 2021: 212-231.
> [2] Chen S, Song Z, Haque M, et al. Nicgslowdown: Evaluating the efficiency robustness of neural image caption generation models. In Proceedings of the IEEE/CVF Conference on Computer Vision and Pattern Recognition (CVPR). 2022: 15365-15374.
> [3] Chen S, Liu C, Haque M, et al. Nmtsloth: understanding and testing efficiency degradation of neural machine translation systems. In Proceedings of the 30th ACM Joint European Software Engineering Conference and Symposium on the Foundations of Software Engineering (ESEC/FSE). 2022: 1148-1160.
> [4] Hong S, Kaya Y, Modoranu I V, et al. A panda? no, it's a sloth: Slowdown attacks on adaptive multi-exit neural network inference. In International Conference on Learning Representations (ICLR), 2021.
> [5] Chen S, Chen H, Haque M, et al. The Dark Side of Dynamic Routing Neural Networks: Towards Efficiency Backdoor Injection. In Proceedings of the IEEE/CVF Conference on Computer Vision and Pattern Recognition (CVPR). 2023: 24585-24594.
> [6] Liu H, Wu Y, Yu Z, et al. SlowLiDAR: Increasing the Latency of LiDAR-Based Detection Using Adversarial Examples. In Proceedings of the IEEE/CVF Conference on Computer Vision and Pattern Recognition (CVPR). 2023: 5146-5155.

---

> > ### Comment · Reviewer_xu4N · 2023-11-22
> > **Acknowledgment**
> >
> > Thanks for your response. I will keep my score.

---

> > > ### Author Response · Authors · 2023-11-22
> > > **Thanks for Reviewer xu4N's Response**
> > >
> > > Thanks for your recognition of this work and all valuable comments. We greatly appreciate your thorough review and valuable feedback throughout this process.

---

### Author Response · Authors · 2023-11-20
**General Response to All Reviewers**

We thank all the reviewers for the valuable comments and helping us to improve the paper. We have updated our paper according to reviewers' suggestions and highlighted these revisions **in blue color** in our revised paper. The revisions are summarized as follows:
- Added the explanation to clarify why the length distribution of our verbose images results in bimodal distribution (two-mode distribution) in Appendix D.
- Added a new discussion about a simple defense strategy that limits the generation length in Appendix G.1.
- Added a new discussion about the model performance of three energy-latency attacks in Appendix G.2.
- Added experimental results of image embedding distance between the original image and the verbose image in Appendix G.3.
- Added experimental results of energy consumption for generating one verbose image in Appendix G.4.
- Added a new discussion to show the results of the standard deviation results in Appendix G.5.

---

### Meta-Review · Area_Chair_mJKF · 2023-12-02

**Metareview:**

This paper proposes an attack that makes a VLM model generate a very long sequence such that its power consumption become much higher. The authors have done a good job during rebuttal. After rebuttal, it received scores of 668.

Initially, two reviewers gave slightly negative scores; after rebuttal, all the reviewers become positive about the paper. Specifically, (1) the paper investigates an interesting problem, which attacks the inference efficiency of VLMs. The proposed solution is relatively simple. (2) The paper is well-written and easy to follow. The figures are presented well. (3) Experiments are convincing. Therefore, the AC would like to recommend acceptance of the paper.

**Justification For Why Not Higher Score:**

This paper proposes an attack that makes a VLM model generate a very long sequence. Generally, the AC agrees that the study is interesting, but also may not be significant enough to be a spotlight paper.

**Justification For Why Not Lower Score:**

After rebuttal, all the authors are positive about the paper, so this paper deserves to be accepted.

---

### Decision · Program_Chairs · 2024-01-16

Accept (poster)